# An integrated machine learning approach delineates an entropic expansion mechanism for the binding of a small molecule to α-synuclein

**Sneha Menon†, Subinoy Adhikari†, Jagannath Mondal\***

Tata Institute of Fundamental Research, Hyderabad, India

## eLife Assessment

This study describes the application of machine learning and Markov state models to characterize the binding mechanism of alpha-Synuclein to the small molecule Fasudil. The results suggest that entropic expansion can explain such binding. However, the simulations and analyses in their present form are **inadequate**.

**\*For correspondence:**
jmondal@tifrh.res.in

†These authors contributed equally to this work

**Competing interest:** The authors declare that no competing interests exist.

**Abstract** The mis-folding and aggregation of intrinsically disordered proteins (IDPs) such as α-synuclein (αS) underlie the pathogenesis of various neurodegenerative disorders. However, targeting αS with small molecules faces challenges due to the lack of defined ligand-binding pockets in its disordered structure. Here, we implement a deep artificial neural network-based machine learning approach, which is able to statistically distinguish the fuzzy ensemble of conformational substates of αS in neat water from those in aqueous fasudil (small molecule of interest) solution. In particular, the presence of fasudil in the solvent either modulates pre-existing states of αS or gives rise to new conformational states of αS, akin to an ensemble-expansion mechanism. The ensembles display strong conformation-dependence in residue-wise interaction with the small molecule. A thermodynamic analysis indicates that small-molecule modulates the structural repertoire of αS by tuning protein backbone entropy, however entropy of the water remains unperturbed. Together, this study sheds light on the intricate interplay between small molecules and IDPs, offering insights into entropic modulation and ensemble expansion as key biophysical mechanisms driving potential therapeutics.

## Introduction

Intrinsically disordered proteins (IDPs) comprise a class of proteins that lack a unique, well-defined three-dimensional structure under physiological conditions. Defying the classical structure-function paradigm (*Wright and Dyson, 2015*; *Babu et al., 2011*), they are involved in crucial cellular processes such as regulation of cell cycle signalling and act as hubs in interaction networks mainly due to their promiscuous binding nature (*Csizmok et al., 2016*; *Wang et al., 2011*; *Dyson, 2016*). Emergent studies present disordered proteins as an important component of biomolecular condensates that play regulatory roles in several key cellular functions (*Alberti et al., 2019*; *Chong and Forman-Kay, 2016*; *Shin and Brangwynne, 2017*; *O'Flynn and Mittag, 2021*).

Considering the broad range of functionality of IDPs, their disregulation and altered abundance can lead to a number of diseases including cancer, diabetes, cardiovascular and neurodegenerative disorders (*Wright and Dyson, 2015*; *Babu et al., 2011*; *Uversky et al., 2008*). In rational structure-based

drug design, exploration of the target proteins is a crucial step based on which the interactions of small molecules are optimized to have stabilising interactions with a structured ligand-binding pocket (*Bronowska, 2011*; *Klebe, 2015*) and is typically applied to folded proteins. IDPs, however, are considered *undruggable* since they exist as an ensemble of conformations that precludes the usage of these traditional drug-design strategies (*Metallo, 2010*). Nevertheless, a few monomeric disordered proteins such as the oncogenic transcription factor c-Myc (*Follis et al., 2008*), Abeta42 (*Ono et al., 2012*; *Zhu et al., 2013*; *Attanasio et al., 2013*; *Ehrnhoefer et al., 2008*; *Heller et al., 2020*),Tau (*Akoury et al., 2013*), PTP1B (*Krishnan et al., 2014*), and osteopontin *Kurzbach et al., 2014* have been successfully targeted by small molecules (*Biesaga et al., 2021*). The IDP of this article's interest, alpha-synuclein (αS) has remained an important drug-target (*Fields et al., 2019*; *Tóth et al., 2014*; *Tatenhorst et al., 2016*; *Cao et al., 2022*) owing to its long-rooted association with neurodegenerative diseases. Often, the early aggregation stages of αS into small oligomers is implicated in the causation of Parkinson's disease (*Winner et al., 2011*; *Emin et al., 2022*; *Stephens et al., 2023*; *Ubbiali et al., 2022*) and a potential therapeutic strategy has been the stabilization of αS in soluble monomeric form. In particular, the small molecule fasudil (focus of the present investigation) has been shown to interact with αS and retard its aggregation (*Tatenhorst et al., 2016*). The anti-aggregative effect of fasudil on αS assembly was tested both in vivo and in vitro and it was found to significantly reduce αS aggregation in both scenarios. Characterisation of the effect of small molecule binding on the ensemble properties of IDPs and the molecular details of the stabilizing interactions can provide the basis for selectivity and can be effectively calibrated to induce the desired effect to alleviate disease conditions. In particular, latest effort by Robustelli and coworkers (*Robustelli et al., 2022*) had produced an unprecedentedly long-time scale (1500 μs) all-atom molecular dynamics (MD) simulation trajectory of a small molecule drug, fasudil with monomeric αS, in which the predicted protein-fasudil interactions were found to be in good agreement with previously reported NMR chemical shift data. However, comparison of the global and local structural features of the αS ensemble in neat water and that in the presence of fasudil (*Robustelli et al., 2022*; see *Figure 1—figure supplements 1–6*) did not indicate a significant difference that is a customary signature of the dynamic IDP ensemble. Characterization of the peculiar nature of IDPs such as αS using the conventional experimental and computational approaches have proven to be challenging. IDPs can be best described by an ensemble of conformations potentially sampled by the protein along with their associated statistical/thermodynamic weights. While biophysical methods can provide an ensemble averaged conformation of an IDP, computer simulations can extensively sample the conformational ensembles of IDPs at atomic resolutions. However, biophysical methods lack the potential to provide an ensemble view and there are inherent limitations of force fields used in computer simulations to model IDPs in agreement with experiments. Integrative approaches have been exploited in studying IDPs as well as small-molecule binding to IDPs (*Bonomi et al., 2016*; *Bottaro and Lindorff-Larsen, 2018*; *Robustelli et al., 2018*; *Gomes et al., 2020*; *Stelzl et al., 2022*). In recent years, Markov State Models (MSMs) have been exploited to elucidate a states-and-rates view of biomolecular dynamics i.e. finding the kinetically relevant states and the transition rates between them (*Prinz et al., 2011*; *Bowman et al., 2009*; *Husic and Pande, 2018*). The first step in the process of building a kinetic model / MSM of any complex biomolecular ensemble is to identify a suitable metric or collective variable (CV) that efficiently captures its conformational space.

CVs such as torsion angles, distances, chemical shifts and so forth, can characterize the complex molecular motion in a high number of dimensions. However, this presents a challenge to effectively analyze trajectories generated from MD simulations. In order to address this, dimension reduction is performed to obtain a set of slow variables that can be further subjected to clustering into kinetically discrete metastable states. Since the high-dimensional input features describing the heterogeneous conformational landscape of IDPs can have non-linear relationships, non-linear dimension reduction techniques are most suitable. Recent advances in the field of machine learning have shown better results in capturing such non-linear relationships (*Adhikari and Mondal, 2023*). In this study, we employed a deep neural network, specifically β-variational autoencoder (*Kingma and Welling, 2013*; *Rezende et al., 2014*) to perform dimension reduction and further utilized this machine-learnt latent dimension for building MSM in order to better understand the conformational landscape of IDPs in presence and in absence of a small molecule. In the current work, to elucidate the small-molecule mediated modulatory effect on αS ensemble, we harnessed deep-learning based β-Variational AutoEncoder (β-VAE)

and Markov State Models (MSM) to dissect αS ensemble in water and in aqueous fasudil solution. A projection of the atomistically simulated conformation ensemble of αS in its apo state and in presence of Fasudil, along their respective latent space describes the conformational landscape, which is further refined by developing individual MSMs. Our result reveals that, compared to the macrostates identified in water, the presence of fasudil led to an increase in the number of metastable states. Characterisation of small-molecule interaction with the protein indicates that fasudil has differential interactions with each of these macrostates. These macrostates exhibit clear distinctions as evident from their $C_\alpha$ contact maps which was further analysed using a convolutional VAE. An entropic perspective at the global (whole protein) and local (residue) level reveals that fasudil binding can potentially trap the protein and the resultant states may be disordered or ordered in nature. The effect is mainly manifested at the backbone level with a decrease in entropy at the fasudil binding hotspots in some states. The observations reported in this study demonstrate how the binding of a small molecule, fasudil, modulates the conformational properties of the disordered protein, αS, manifested as a change in the backbone entropy of the fasudil binding hotspots.

## Results and discussion
### Modulation of free energy landscape of αS in presence of fasudil

Simulations of αS monomer in the presence of the small molecule fasudil and 50 mM NaCl have been reported in a previous study by *Robustelli et al., 2022*. This study showed no large-scale differences between the bound and unbound states of αS. In our study, in order to compare the αS-fasudil ensemble with an apo ensemble at same salt concentration, we spawned multiple all-atom MD simulations of αS in neat water and 50 mM NaCl from multiple starting conformations to generate a cumulative ensemble of ~62 μs.

In a clear departure from the classical view of ligand binding to a folded globular protein, the visual change in αS ensemble due to the presence of small molecule is not so strikingly apparent. In order to understand the underlying ensemble modulatory effect of the small-molecule binding events, the complex and fuzzy conformational ensemble of αS (see *Figure 1a*) needs to be delineated using a suitable spatial and temporal decomposition into its key sub-ensembles or metastable states. This prompted us to analyse the two ensembles using the framework of MSM. As a suitable input feature to build the MSM, we estimated the set of inter-residue Cα pairwise distances, as this feature largely incorporates the conformational space of αS. In a standard MSM analysis, this high-dimensional input feature requires further processing such as dimensionality reduction, before discretisation using clustering for the construction of an MSM.

We started our investigation by searching for an optimized representation of high-dimensional and heterogeneous ensemble of monomeric αS conformation in presence of fasudil. As the datasets, particularly in the context of IDPs, are more complex and large in size, the need for effective dimensionality reduction techniques becomes more conspicuous. Dimension reduction using principal component analysis, independent component analysis (*Hyvärinen, 2013*), singular value decomposition (*Klema and Laub, 1980*) and linear discriminant analysis (*Fisher, 1936*) linearly transform the high dimensional data into a lower dimensional manifold. However, non-linear methods such as kernel PCA (*Schölkopf et al., 2005*), multidimensional scaling (*Kruskal and Wish, 1978*), isomap (*Tenenbaum et al., 2000*), fastmap (*Faloutsos and Lin, 1995*), locally linear embedding (*Roweis and Saul, 2000*) etc *Sumithra and Surendran, 2015* have outperformed such linear transformation methods and have been used for free energy calculations (*Das et al., 2006*) and determining timescales to capture slow conformational changes (*Schwantes and Pande, 2015*; *Wehmeyer and Noé, 2018*). In the present work, we opted to draw inspiration from artificial deep neural network based framework to employ model-agnostic and mostly unsupervised non-linear approach to derive optimized non-linear latent feature space that can be employed for statistical state-space decomposition of IDP such as αS.

Over the recent years, unsupervised machine learning algorithms such as autoencoders have been effective as a dimension reduction tool, owing to non-linear activations of the neurons that allows them to capture complex relationships and non-linear patterns in the high dimensional data (*Hinton and Salakhutdinov, 2006*; *Adhikari and Mondal, 2023*). This high dimensional input is represented into a lower dimensional latent space, which is then used to reconstruct the input. However, the latent space is 'deterministic', that is it provides only a fixed mapping of the input to latent space, which might

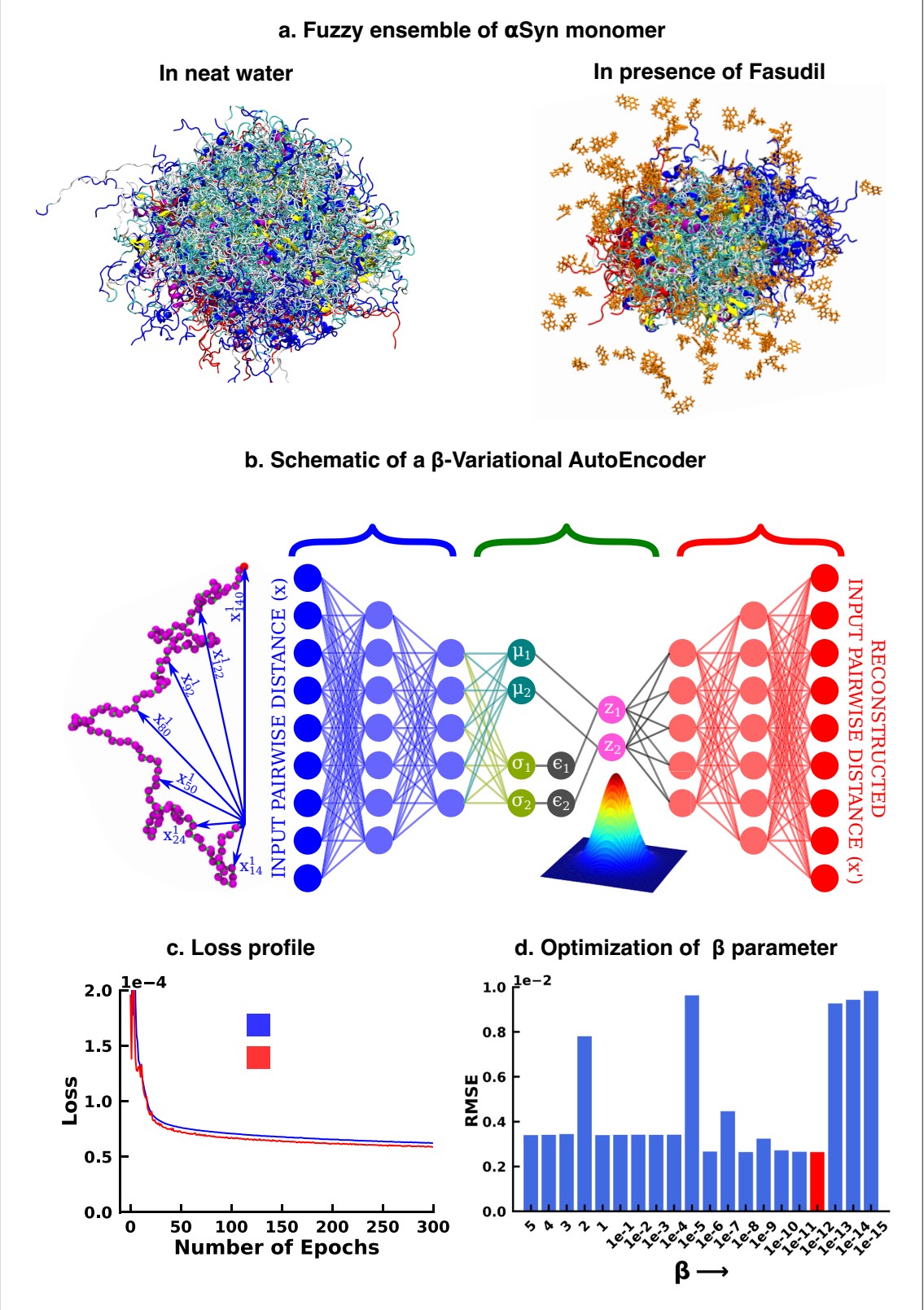

**Figure 1.** Deep neural network illustrating dimension reduction through β-Variational AutoEncoder. (**a**) Conformational ensemble view of αS and αS-Fasudil ensemble, (**b**) A schematic of β-Variational Autoencoder (β-VAE), (**c**) Training and validation loss for β-VAE and (**d**) RMSE as a function of the β parameter. The red-annotated β parameter was used for the investigation.

*Figure 1 continued on next page*

*Figure 1 continued*

The online version of this article includes the following figure supplement(s) for figure 1:

**Figure supplement 1.** Distribution of the radius of gyration (Rg) of the αS and αS-Fasudil ensembles.

**Figure supplement 2.** Distribution of end-to-end distance for the αS and αS-Fasudil ensembles.

**Figure supplement 3.** Distribution of SASA for αS and αS-Fasudil ensembles.

**Figure supplement 4.** Residue wise percentage secondary structure of helical nature in the αS and αS-Fasudil ensembles.

**Figure supplement 5.** Residue wise percentage secondary structure of sheet nature in the αS and αS-Fasudil ensembles.

**Figure supplement 6.** Residue wise percentage secondary structure of coil nature in the αS and αS-Fasudil ensembles.

**Figure supplement 7.** Each blue diamond label represents the RMSD (nm) w.r.t the starting structure of aSyn-fasudil simulation.

**Figure supplement 8.** Fraction of helix content in the starting structures of the αS-Fasudil and αS simulations.

**Figure supplement 9.** Fraction of β-sheet content in the starting structures of the αS-Fasudil and αS simulations.

**Figure supplement 10.** Fraction of coil content in the starting structures of the αS-Fasudil and αS simulations.

**Figure supplement 11.** Total SASA of the starting structures of the αS-Fasudil and αS simulations.

**Figure supplement 12.** Residue wise SASA of the starting structure of the αS-Fasudil (red square) and αS (blue squares) simulations.

not account for variations or uncertainty present in the data. Consequently, this could generate an inaccurate result for a point that does not exist in the latent space. Thus, when projecting new, similar data onto the latent space, a deterministic autoencoder will map similar inputs to nearly identical points in the latent space, potentially lacking the diversity and capturing only a single mode of the data distribution. In contrast, the latent space of a Variational Autoencoder (VAE) is 'probabilistic' in nature, thereby avoiding this limitation of autoencoders. The technical details have been documented in the Method section. However, here we provide a brief rationale for the choice of the β-VAE model.

Variational autoencoder (VAE) (*Kingma and Welling, 2013*; *Rezende et al., 2014*; *Figure 1b*) is also an unsupervised machine learning algorithm, which although is closely related to autoencoder, it is inextricably linked with variational bayesian methods. A VAE consists of an encoder network and a decoder network, much like a traditional autoencoder. The encoder takes input data and maps it to a probabilistic distribution in the latent space, while the decoder reconstructs the input data from samples drawn from this distribution. The probabilistic nature of the latent space is a result of applying variational inference, which aims to estimate the actual posterior using an approximate distribution, parameterized by a mean vector and a variance vector. This estimation is achieved by minimizing the Kullback-Leibler (KL) divergence (also known as relative entropy or information gain) between the two distributions. As a result, instead of producing a single point in the latent space for each input, the encoder outputs a distribution. The latent variable is then sampled from this distribution, which is then used by the decoder to reconstruct the input. This makes the latent space continuous, thereby improving its ability to interpolate novel, unseen data. The VAE model can be further improved by enhancing the probability of generating an actual data while keeping the distance between the true and the posterior distribution under a small threshold. This results in the weighting of the KL divergence term in the loss function. The factor determining this weight is represented by β and hence the name β-VAE, which is used in this study. The implementation of the β-VAE is made publicly available in our group's Github page (copy archived at *JMLab-tifrh, 2024*). The loss function of β-VAE is given as,

$$\text{Loss}_{VAE} = \text{Reconstruction Loss} + \beta \times \text{KL Divergence} \tag{1}$$

The β-VAE model was trained over a number of β values ranging between 5 and $1 \times 10^{-15}$. A β value of $1 \times 10^{-12}$ was chosen as this value gave us the minimum RMSE between the original and the reconstructed data(see *Figure 1d*). This model was further used to project the αS data in its apo state for further analysis.

We have also computed the VAMP2 score after dimension reduction of Cα pairwise distance using a linear dimension reduction method namely time-lagged independent component analysis (tICA) and compared it with the score obtained from the latent dimension of β-VAE. We found that the VAMP2 score obtained from β-VAE was consistently higher than tICA derived dimensions for the αS-Fasudil ensemble, thereby indicating that β-VAE would help identifying relatively slower CVs (see

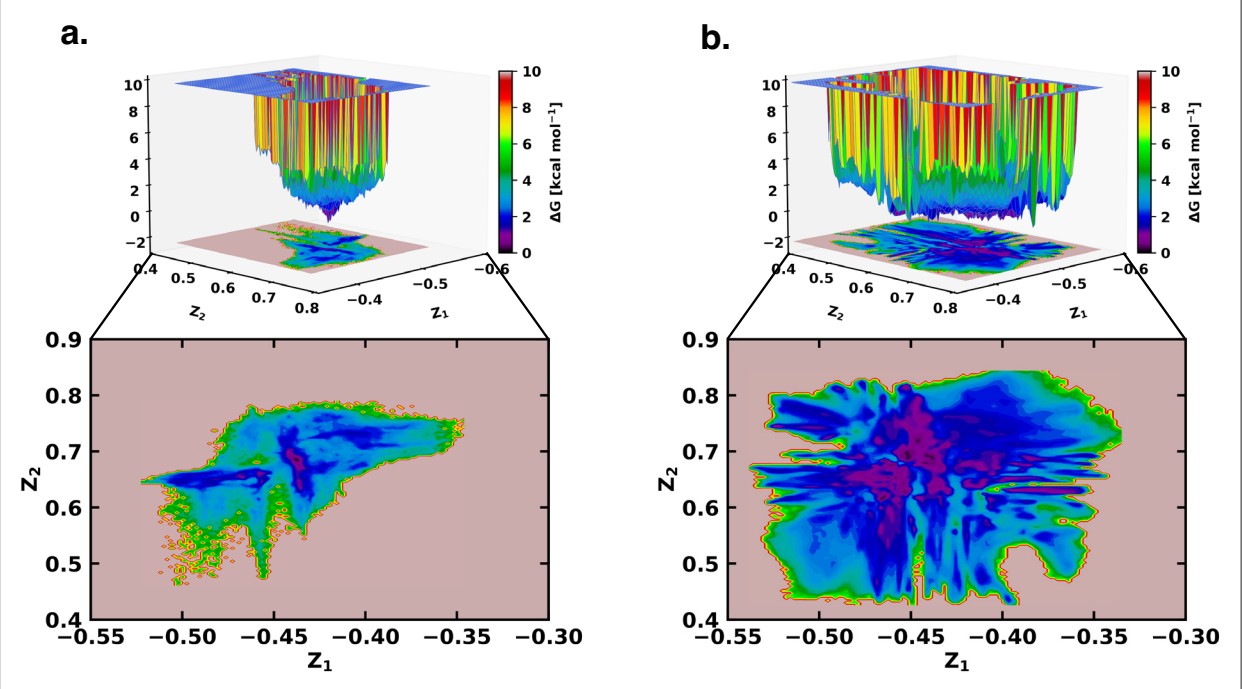

**Figure 2.** Free energy landscape using the latent space of β-VAE indicates greater conformational space sampled by αS in presence of fasudil. The free energy landscape of (**a**) αS and (**b**) αS-Fasudil system as determined from the latent dimensions of β-VAE.

Figure 3—figure supplements 1–2) as compared to tICA and hence better at representing slower dynamical processes in the αS and αS-Fasudil simulations.

In particular, as described in detail in the Methods section, we trained a VAE model using a large array of inter-residue pairwise distances using the β-VAE architecture. Our attempt to reconstruct the large-dimensional input feature via a β-VAE resulted in compression of the 1.5 ms MD simulation data into a two-dimensional latent space. The loss function steadily decreased and eventually plateaued providing a model optimized for this system (see *Figure 1c*). The rugged nature of the free energy landscape of αS, is evident from its projection along the latent space dimension of the simulated trajectory either in aqueous media (see *Figure 2a*) or in presence of fasudil (see *Figure 2b*). The free energy landscape represents a large number of spatially close local minima representative of energetically competitive conformations inherent in αS, which is also the source of the conformational heterogeneity and a hallmark of an IDP(see *Figure 2a–b*). Next we wanted to compare the changes in the conformational landscape of the apo αS conformations. Subsequent projection of the apo state on the low-dimensional subspace also brought out a spatially heterogeneous landscape.

However, a close comparison of free energy landscapes indicates that a set of local minima representing conformations of αS are less clustered and appear in small patches in the apo state. Moreover, a relative comparison of the two free energy landscapes suggests that αS spans a significantly larger space in presence of fasudil even in this low dimensional subspace, hinting at an expansion of conformational repertoire of this IDP in presence of the small molecule (see *Figure 2a–b*). Individual, one-dimensional projection of the conformational landscape along each of the two latent dimensions reveals that the presence of the small molecule would shift part of the conformational landscape to a distinct location, suggesting that fasudil would modulate the existing conformational ensemble as well as would create new conformational space. For a more discrete and clearer state-space decomposition of the conformational ensemble of αS (in apo or in ligand-bound form), we attempt to enumerate the optimum number of distinct metastable macrostates with non-negligible equilibrium populations by constructing MSM as explained in the following section.

## MSM elucidates distinct binding competing states of αS in presence of the small-molecule drug

The two-dimensional feature obtained from β-VAE was used as the input feature for the MSM estimator provided by PyEMMA (*Scherer et al., 2015*). A flowchart of the implemented protocol is drawn in *Figure 3a*. A geometric clustering approach, *k*-means, was used to discretise the ensemble. Analysis of the implied time scales as a function of lag-time predicted a Markovian model of αS in water with three kinetically separated states (see *Figure 3d* for relative population). On the other hand, a similar analysis of αS in presence of fasudil (*Figure 3c*) indicated a clear increase in the number of spatially and temporally resolved conformational metastable states. Thus, a three-state MSM was built for the αS ensemble in water whereas a six-state model was built for the αS-fasudil ensemble. Hereafter, we will refer to the macrostates created in presence of fasudil prefixed with FS and those generated in water prefixed with MS. A bootstrapping analysis was performed to estimate the mean and standard deviations of the equilibrium populations of these metastable states (*Figure 3d*).

The MSM was also further validated for Markovianity i.e. ability to predict estimates at longer time scales using the Chapman-Kolmogorov (CK) test, as depicted in *Figure 3—figure supplement 3* for αS-Fasudil simulation.This test further validates the quality of the model that it estimates long-time scale behavior with reasonable accuracy. The MSM analyses essentially indicate that the small-molecule binding interactions with αS led to sampling or generation of additional conformational states than the states populated in the apo αS ensemble.

A pertinent question arises: Are the appearances of newer states in αS in presence of fasudil a result of the presence of longer trajectory in fasudil solution (1500 μs) than that of the ensemble populated in neat water (62 μs)? We verified this by building an MSM for 60 μs of data of fasudil system using 10 μs to 70μs segment of the trajectory (see *Figure 3—figure supplement 4*). We find that the MSM built using lesser data (and same amount of data in the αS simulation) also indicated the presence of six states of αS in presence of fasudil, as was observed in the MSM of the full trajectory. We compared the $R_g$ distribution of this small chunk with the αS ensemble and found to be in good agreement (see *Figure 3—figure supplement 5*). Addidtionally, we selected a chunk spanning from 966 μs to 1026 μs and observed a similar distribution of $R_g$ (see *Figure 3—figure supplement 6*). Moreover, we built a MSM on this new chunk and identified the presence of six states (see *Figure 3—figure supplement 7*). Together, this exercise invalidates the sampling argument and suggests that the increase in the number of metastable macrostates of αS in fasudil solution relative to that in water is a direct outcome of the interaction of αS with the small molecule.

Another aspect that we evaluated was in the choice of the number of latent dimensions of β-VAE. Even though an increase in the number of latent dimensions may make the model more accurate, this can also result in overfitting. The model can simply memorize the pattern in the data instead of generalizing them. A higher dimensional latent space is also more difficult to interpret; therefore, we chose two dimensions. Nevertheless, we trained another β-VAE with four neurons on the latent space and built a MSM by choosing the appropriate number of microstates (see *Figure 3—figure supplement 8*). The implied timescales (see *Figure 3—figure supplement 9*) indicate the presence of six states which is consistent with the model with two latent dimensions.

## Structural characterisation of metastable states of αS monomer in presence of fasudil

We characterised the residue-wise intramonomer contact maps of the metastable states to identify the differences in the interaction patterns in these states that can be attributed to interactions of fasudil with the monomer. The average inter-residue contact probability maps for each of the metastable states populated in the presence of fasudil and in water are depicted in *Figures 4 and 5*, respectively. The residue-wise percentage secondary structure in each of the macrostates are presented in *Figure 4—figure supplement 1*, *Figure 5—figure supplement 1*, respectively. In state FS1, antiparallel β-sheet interactions are formed within the N-terminal region. This β-sheet network also includes long-range interactions of the H2 region of the N-terminus with the negatively charged C-terminal region. Residues 70–80 in the hydrophobic NAC region are also involved in antiparallel β-sheet interactions with residues 120–130 in the C-terminal region. Short parallel β-sheet interactions also exist between the NAC and C-terminal residues with the N-terminal residues. The residues in the C-terminus exhibit 40–90% of β-sheet propensity (see *Figure 4—figure supplement 1*). Long

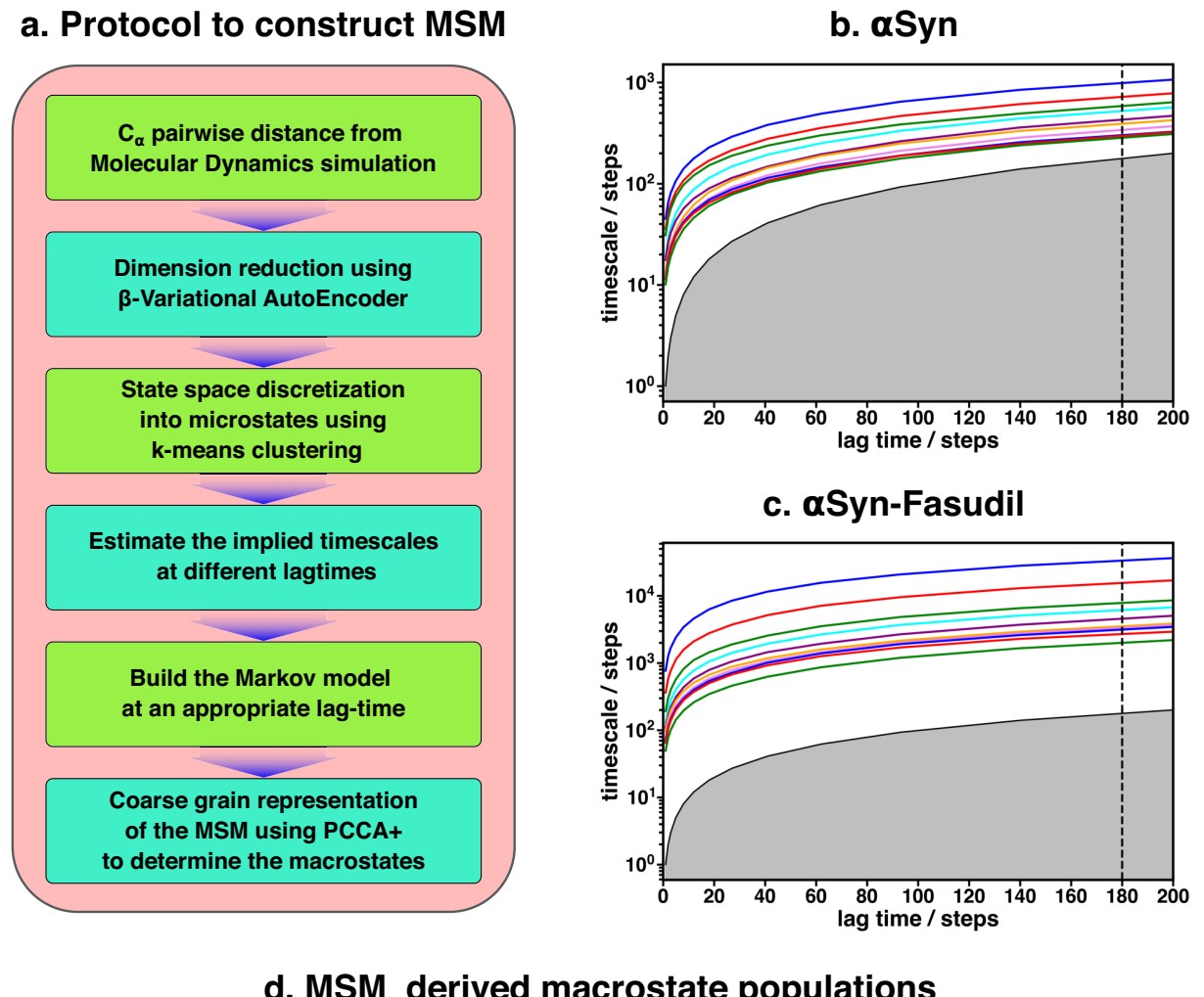

**Figure 3.** Markov state model captures three and six states of αS and αS-Fasudil system, respectively. (**a**) A flowchart of the process of building a Markov State Model (MSM).(**b, c**) Implied timescales plot of the αS and αS-Fasudil systems. (**c**) Macrostate populations of the 3-state and 6-state MSM of the αS and αS-Fasudil systems. Bootstrapping was used to estimate the mean and standard deviations of the macrostate populations.

The online version of this article includes the following figure supplement(s) for figure 3:

**Figure supplement 1.** Comparison of VAMP-2 score from VAE and tICA of pairwise distances, reduced to two dimensions for the αS-Fasudil simulation.

**Figure supplement 2.** Comparison of VAMP-2 score from VAE and tICA of pairwise distances, reduced to two dimensions for the αS simulation.

**Figure supplement 3.** The Chapman-Kolmogorov test performed for the six state Markov State Model of the αS-Fasudil ensemble.

**Figure supplement 4.** The implied timescale (ITS) plot αS-Fasudil simulation for the block of 10 μs to 70 μs.

*Figure 3 continued on next page*

*Figure 3 continued*

**Figure supplement 5.** Distribution of Rg of the αS ensemble and the 60 µs slice (10 µs-70 µs) of the αS-Fasudil simulation trajectory.

**Figure supplement 6.** Distribution of Rg of the αS ensemble and the 60 µs slice (966 µs to 1026 µs) of the αS-Fasudil simulation trajectory.

**Figure supplement 7.** The implied timescale (ITS) plot αS-Fasudil simulation for the block of 966 µs to 1026 µs.

**Figure supplement 8.** VAMP-2 score as a function of the number of microstates in for the αS-Fasudi ensembles for a latent space of four dimensions.

**Figure supplement 9.** ITS plot of the αS-Fasudi simulation using the four dimension of the VAE.

**Figure supplement 10.** VAMP-2 score as a function of the number of microstates in for the αS-Fasudil ensemble.

**Figure supplement 11.** VAMP-2 score as a function of the number of microstates in for the αS ensemble.

range interactions between N-terminus and C-terminus are relatively diminished in state FS2. This state has multiple parallel β-sheet interactions between short stretches of residues in the NAC region and C-terminal region with the N-terminal region. In addition, antiparallel β-sheet interactions are also present in NAC:N-terminus, NAC:C-terminus and within the N-terminal region. Residues 8–13 in the N-terminus and 86–92 in the NAC region exhibit significant helicity. State FS3 is characterised by ~20% of β-sheet interactions across the entire protein. Hydrophobic and polar, uncharged residues in the range 61–80 in the NAC region are involved in β-sheet interactions with the N-terminus(residues 1–10) and C-terminus(100–120). Moreover, β-sheet interaction networks are also present within the NAC and N-terminal regions. The C-terminal region 100–120 forms long-range interactions with residues 1–10 in the N-terminal region. Extensive short and long-range β-sheet networks are prevalent in state FS4. The entire NAC region forms antiparallel β-sheet network with the N-terminus. The C-terminusforms parallel β-sheets with the NAC and N-terminus. In the most populated state FS5, the key interactions are β-sheets formed between the oppositely charged termini. The NAC region also exhibits $\alpha$-helical propensity. Finally, in state FS6, the NAC region (residues 60–80) forms antiparallel β-sheet network with the residues 30–60 of the N-terminal segment as well as the C-terminal region 100–120. These residues have ~40% β-sheet propensity.

On the other hand, the metastable states that are populated in neat water are distinct from those formed in the presence of fasudil. In state MS1, the NAC region residues 61–80 interacts with the N-terminal region by both parallel and antiparallel β-sheet interactions. The β-sheet network of NAC also incorporates interactions with residues 100–130 in the C-terminal region. These residues have ~60% β-sheet propensity (see *Figure 5—figure supplement 1*). State MS2, the most populated macrostate in neat water, is characterised by β-sheet interactions of the NAC region with H2 region (residues 30–60) of the N-terminus. Furthermore, the N and C-termini also form β-sheet interactions. State MS3 mainly has long-range interactions as β-sheets between the N and C-terminal regions. Moreover, a β-sheet network also exists within the NAC region and NAC:N-terminal (H2) interface. The residues in the NAC region have ~50% β-sheet propensity in this macrostate.

We note that the differences in the structural features that is intra-protein contacts of the macrostates arise from the interactions of fasudil with the protein residues. For instance, in state FS2 in which fasudil interacts with multiple sites across the entire length of the protein, the secondary structure properties of this state are dictated by β-sheets formed by short stretches of residues rather than an extensive network. Furthermore, owing to the relatively stronger small-molecule binding interactions with hydrophobic residues in the N-terminus, the long-range interactions of the oppositely charged terminal regions are suppressed. Similarly, in state FS4, since fasudil primarily interacts with residues 120–140 of the C-terminusand other interactions are transient, the conformational properties of this metastable state is marked by a network of β-sheets across the rest of the protein. Essentially, small-molecule binding interactions via stabilisation of various intra-protein interactions modulate the conformational properties of αS.

## Mapping the macrostates in αS and αS-Fasudil ensembles using denoising convolutional variational autoencoder

To ascertain the distinctions among these contact maps for the αS and αS-Fasudil system, we chose to filter the essential features and patterns by representing it in a lower dimensional space. Initially, this was attempted using a Convolutional Variational AutoEncoder (CVAE) with the contact maps of αS and αS-Fasudil macrostates as inputs. The model was initially trained using the six contact maps of αS-Fasudil system as a test case. However, the model displayed a tendency to overfit by learning an

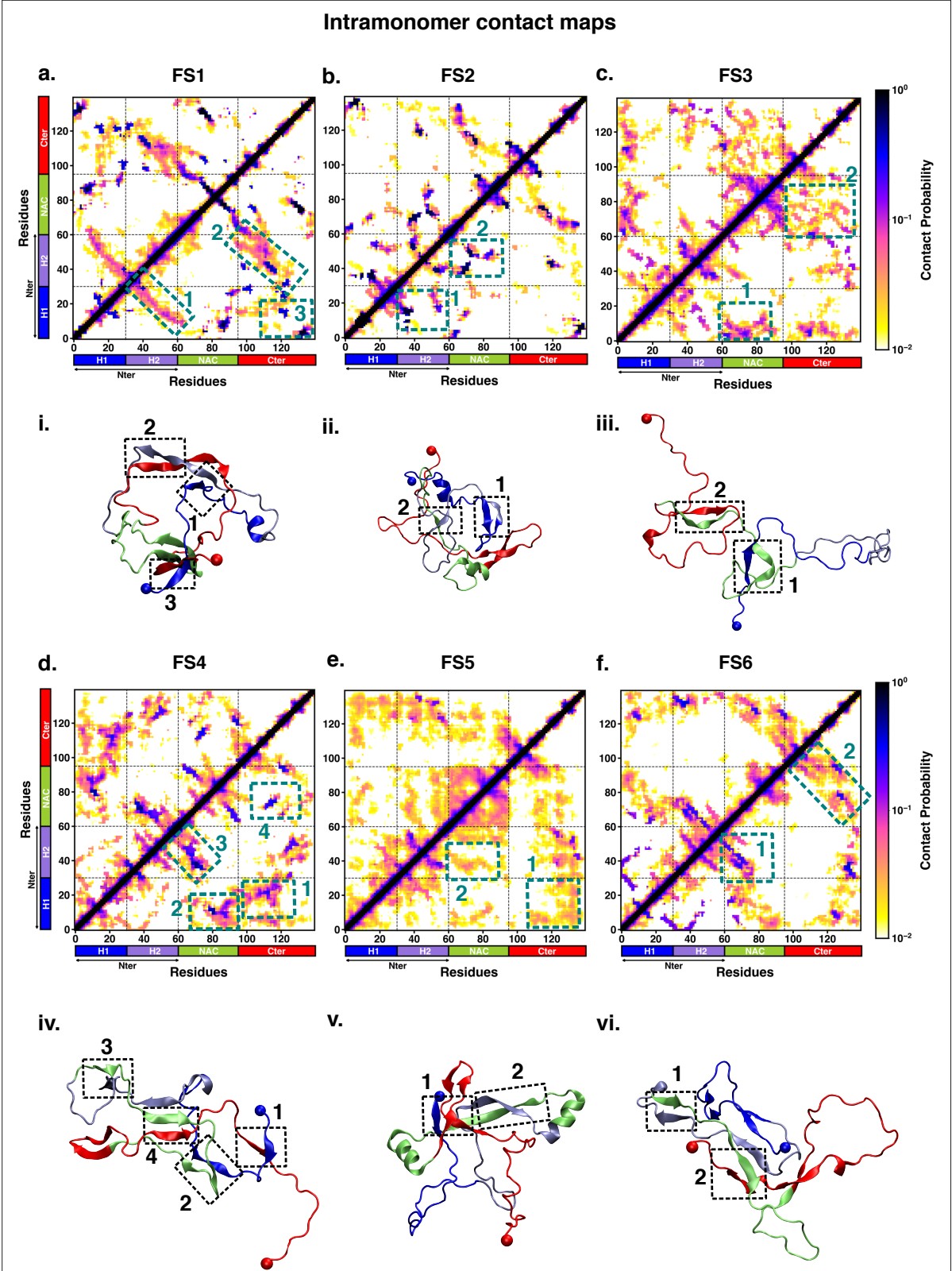

**Figure 4.** Intrapeptide residue-wise contact map of the macrostates of αS-Fasudil system indicates that the six states represent distinct ensembles. Intrapeptide residue-wise contact probability maps of (**a**) six macrostates (FS1 to FS6) in the presence of fasudil. A contact is considered if the Cα atoms of two residues are within a distance of 8Å of eachother. Axes denote the residue numbers. The color scale for the contact probability is shown at the extreme right of each panel of maps. The color bar along the axes of the plots represents the segments in the αS monomer. Specific contact regions are

*Figure 4 continued on next page*

*Figure 4 continued*

marked by boxes and numbered. These contacts are illustrated by representative snapshots and the corresponding contacts are similarly marked and numbered.

The online version of this article includes the following figure supplement(s) for figure 4:

**Figure supplement 1.** Residue-wise percentage secondary structure estimated for each of the six macrostates of αS monomer simulated in the presence of fasudil.

identity function between the input and output, as evident from the large reconstruction error of the test dataset (see *Figure 6—figure supplement 1a*). This overfitting issue was addressed by training a Denoising CVAE (DCVAE), where the contact map was intentionally corrupted with noise prior to training. This approach successfully mitigated the problem of overfitting as indicated by the loss profile of the test dataset (see *Figure 6—figure supplement 1b*). An implementation is provided in GitHub (copy archived at *JMLab-tifrh, 2024*).

The denoising performance was measured by calculating the structural similarity index measure (SSIM) between the original and denoised contact map. SSIM measures the similarity between two images as perceived by the human eye by considering the luminance, contrast and structure of the image. SSIM values can range from –1–1, which represents perfect anti-correlation and perfect similarity, respectively (*Hore and Ziou, 2010*). The SSIM values is tabulated in *Figure 6*. We find that the SSIM values for all the metastable states of αS and αS-Fasudil system are close to 1 (see *Figure 6*), thereby indicating that the denoised contact map as obtained from the trained DCVAE model is very close to the quality of the original contact map (see *Figure 6—figure supplements 2–4*). This is also supported by the high Peak signal-to-noise ratio (PSNR). PSNR is the ratio of the maximum possible power of a signal to the power of corrupting noise that measures the fidelity of a reconstructed image compared to its original in decibels (dB) (*Hore and Ziou, 2010*). A higher value of PSNR indicates smaller difference between the original and denoised contact map (see *Figure 6*). Upon visualizing the latent space, it became evident that the contact map of the metastable states of αS and αS-Fasudil occupy distinct locations within the latent space, which indicates that the DCVAE model effectively captured the distinct patterns among the metastable states.

## Fasudil exhibits conformation-dependent interactions with individual metastable states

We next sought to understand the specificities of the interactions of fasudil with the different metastable states of αS. Accordingly, we calculated the contact probabilities between fasudil and each αS residue. A contact between fasudil and a protein residue is considered to be present when the minimum distance between any heavy atom of fasudil and a protein residue is less than 0.6 nm. This is calculated for the entire population of each macrostate to determine the contact probabilities. The αS-Fasudil contact maps for the six macrostates are presented in *Figure 7a* and snapshots of the interactions are presented in *Figure 7b*.

The residue-interaction profiles of each of the six individual macrostates of αS with fasudil revealed distinct motifs of interaction across the length of the protein for the different macrostates (*Figure 7*). Previousinvestigations *Robustelli et al., 2022*; *Tatenhorst et al., 2016* have reported dominant fasudil-αS interaction via the C-terminal region of the protein while it weakly interacts with the entire αS sequence. In state FS1, fasudil dominantly interacts with residues 133–139 of the C-terminal region, which consists of negatively charged residues D135 and E137 as well as the polar, neutral tyrosine residues (Y133 and Y136). In addition, there are also non-specific interactions of fasudil with hydrophobic residues in the NAC region (residues 71–74). In state FS2, fasudil interactions are spread across the entirety of the protein. These comprise hydrophobic residues in the N-terminal region, neutral polar residues (62–66) in the NAC region and residues 130–136 in the C-terminal region. Fasudil interacts strongly with the NAC region consisting of hydrophobic and polar, neutral residues in state FS3. Interactions also prevail with the negatively charged C-terminal region and residues 1–6 of the N-terminal region. While interacting non-specifically with residues in the amplipathic N-terminal region of state FS4, fasudil preferentially interacts with residues 121–140 of the C-terminal region via charge-charge and π-stacking interactions with the negatively charged residues (D135, E137, E139) and tyrosine side chains (Y125, Y133, Y136). In the most populated state FS5, fasudil interacts weakly with

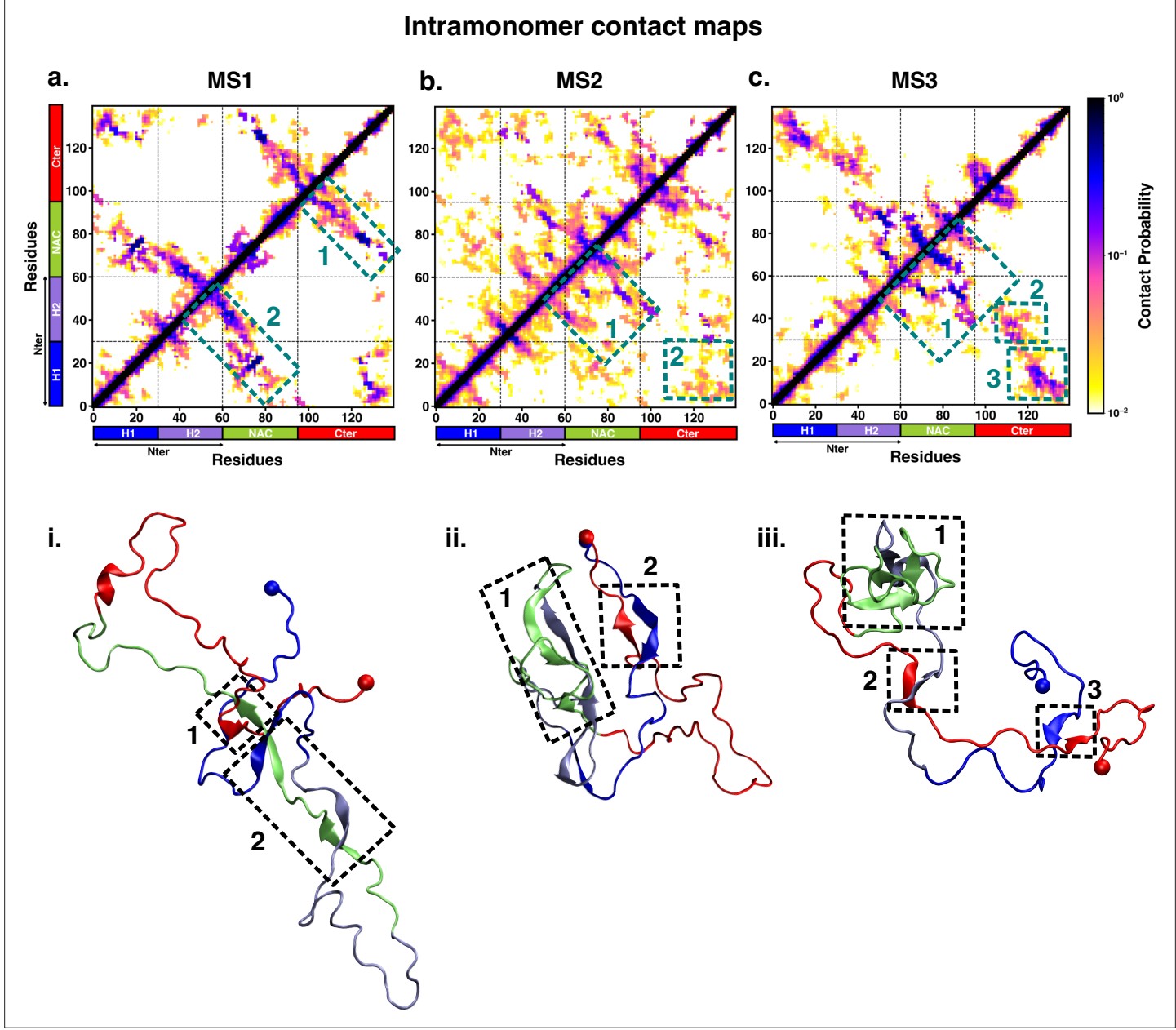

**Figure 5.** Intrapeptide residue-wise contact map of the macrostates of αS system indicates that the three states represent distinct ensembles. Intrapeptide residue-wise contact probability maps of the three macrostates (MS1 to MS3) in neat water. A contact is considered if the $C_\alpha$ atoms of two residues are within a distance of 8Å of eachother. Axes denote the residue numbers. The color scale for the contact probability is shown at the extreme right. The color bar along the axes of the plots represents the segments in the αS monomer. Specific contact regions are marked by boxes and numbered. These contacts are illustrated by representative snapshots and the corresponding contacts are similarly marked and numbered.

The online version of this article includes the following figure supplement(s) for figure 5:

**Figure supplement 1.** Residue-wise percentage secondary structure estimated for each of the three macrostates of αS monomer simulated in neat water.

the hydrophobic residues and glutamic acid residues while interactions with the C-terminal region 107–140 comprising of tyrosines and acidic amino acids are dominant. Lastly, in state FS6, the strongest interactions of fasudil comprise hydrophobic and polar residues in the range 118–121. There are also weak interactions with hydrophobic residues in the N-terminus(11–16, 38, 39) and NAC regions (71–74, 80–95).

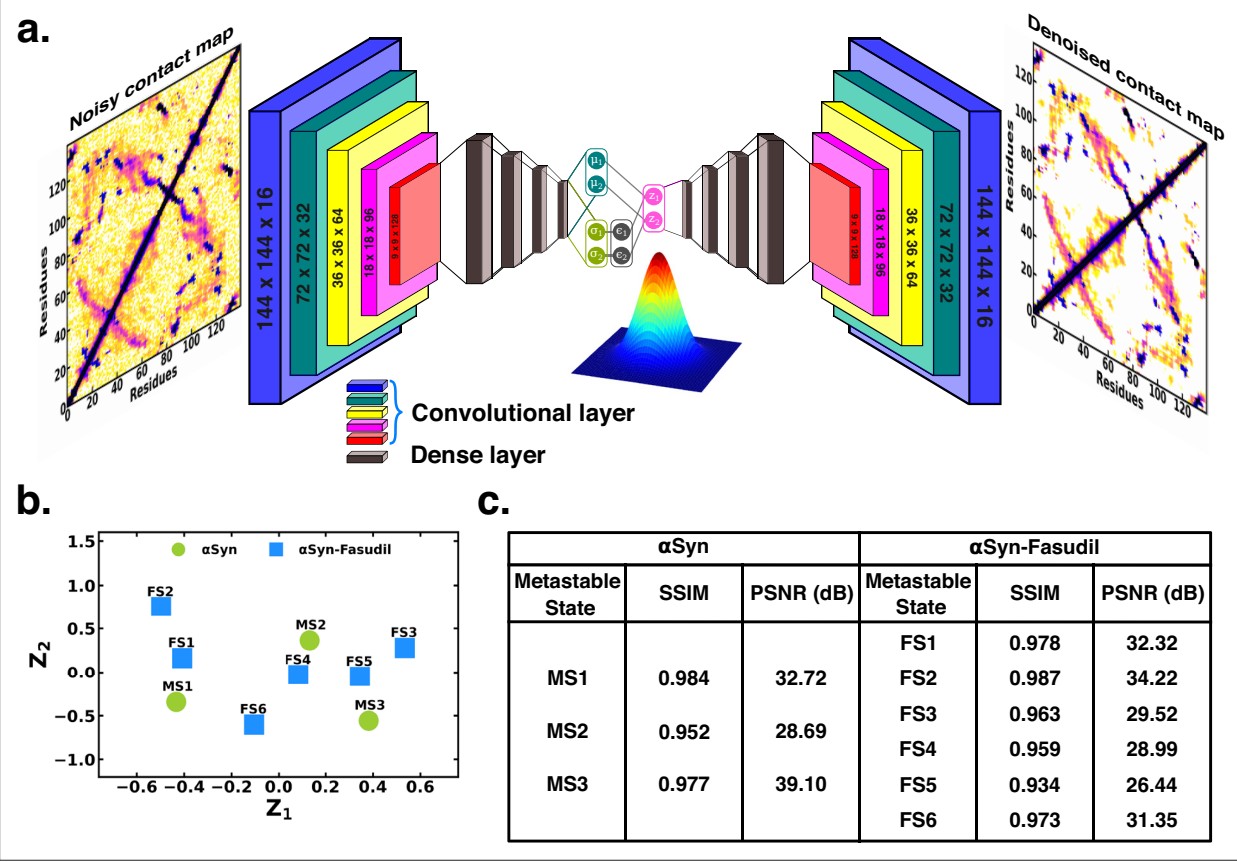

**Figure 6.** Projection of contact maps of αS and αS-Fasudil macrostates using denoising convolutional VAE. (**a**) Schematic of denoising convolutional variational autoencoder (**b**) Latent space of the contact map of αS and αS-Fasudil metastable states. (**c**) SSIM and PSNR values for αS and αS-Fasudil metastable states.

The online version of this article includes the following figure supplement(s) for figure 6:

**Figure supplement 1.** Loss profile of (**a**) CVAE indicating overfitting (**b**) DCVAE indicating no overfitting.

**Figure supplement 2.** Denoised output of the contact map of the αS macrostates.

**Figure supplement 3.** Denoised output of the contact map of the αS-Fasudil macrostates FS1, FS2, and FS3.

**Figure supplement 4.** Denoised output of the contact map of the αS-Fasudil macrostates FS4, FS5, and FS6.

*Robustelli et al., 2022* have reported that in the bound state, fasudil interactions with αS are favorable charge-charge and π-stacking interactions that form and break in a mechanism, they term as *dynamic shuttling*. The stacking interactions arise from the favourable orientation of fasudil's isoquinoline ring and the aromatic ring in the side chain of tyrosine residue. Analysis of the time-series of the formation of different intermolecular interactions indicated the formation of charge-charge and π-stacking interaction with residues Y125, Y133, and Y136 and the shuttling among these interactions causes fasudil to remain localized to the C-terminal region. Notably, we observe that each of the metastable, while preferentially interacting with the C-terminus also has distinct yet significant interactions with hydrophobic residues in the N-terminus and NAC regions.

## Entropic signatures of small molecule binding

The overall analysis of the conformational ensemble of αS in presence of fasudil demonstrates that small-molecule binding substantially modulates the ensemble of αS. Our results indicate that the interaction of fasudil with αS residues governs the structural features of the protein. Estimation of the metastable states of αS shows that compared to aqueous conditions in which three metastable states are identified, the ensemble populated in the presence of fasudil in solution contains six metastable states. Since IDPs are described as an ensemble unlike folded proteins with a fixed structure, the

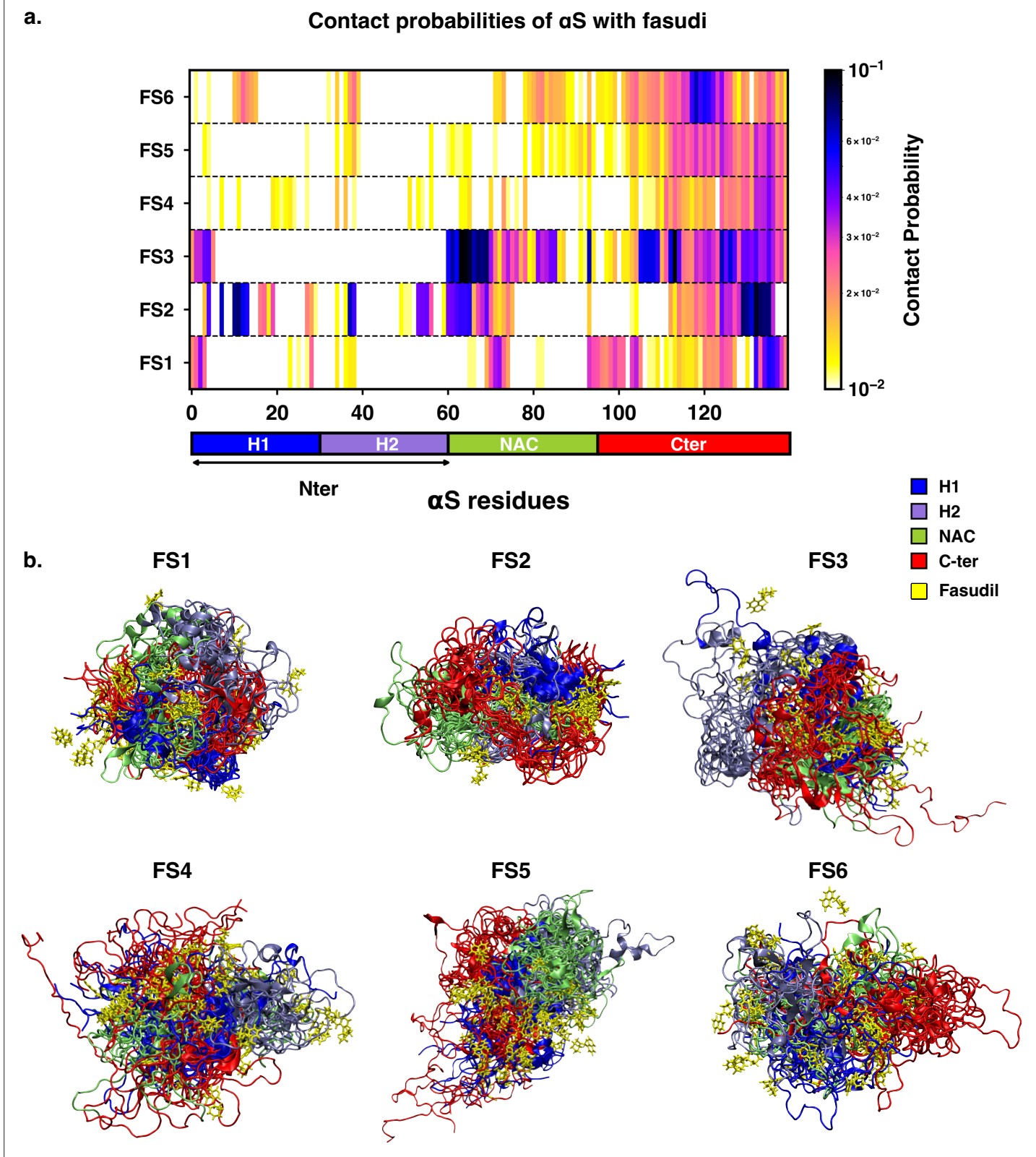

**Figure 7.** Contact map between fasudil and αS, indicates that fasudil interacts with distinct regions of αS in the six macrostates. (**a**) Contact probability map of residue-wise interaction of fasudil with αS.(**b**) Overlay of representative conformations from the six macrostates (FS1 to FS6) along with the bound fasudil molecules. The conformations are colored segment-wise as shown in the legend. The fasudil molecules, in licorice representation, are colored yellow.

effect of small molecules binding on disordered proteins or regions, is manifested as modulation of the whole ensemble by shifting the population of the states leading to a change in its conformational entropy (*Heller et al., 2018*; *Heller et al., 2015*; *Flock et al., 2014*). The common mechanism that is generally postulated is the *entropic collapse* or *folding upon binding* model. (*Dyson and Wright, 2002*) It describes the disorder-to-order transition that IDPs typically undergo upon interacting with its biological target. This transition leads to the shift in population to a more ordered, predominant state causing a loss of conformational entropy that is compensated by a corresponding enthalpic gain. Conversely, small-molecule binding may also lead to *entropic expansion* in which newer conformational states are populated that are otherwise undetectable (*Heller et al., 2015*). Additionally, in cases where there is no gross change in the conformational entropy and the binding is considered *fuzzy* (*Bonomi et al., 2016*), referred to as *isentropic shift*. The observations reported in the present investigation for αS monomer plausibly conform with the entropy-expansion model (*Heller et al., 2018*). According to this model, the interactions of small molecules with IDPs can promote the exploration of conformations with discernible probability, thereby expanding the disorderedness (conformational entropy) of the ensemble (*Heller et al., 2018*; *Heller et al., 2015*; *Heller et al., 2020*; *Löhr et al., 2022*). The small molecule interaction with αS operates transiently, and not strongly as seen in structured proteins. These transient local interactions stabilise the conformations that are inaccessible in the absence of the small molecule.

To gain deeper insights into the entropic effect of small-molecule binding to αS, we further estimated protein conformational entropy as well as water (solvent) entropy. We evaluated the entropy of water using the 2PT (*Lin et al., 2003*; *Lin et al., 2010*) method. First, we determined the molar entropy of pure water that yielded an entropy value of 55.5 J mol$^{-1}$ K$^{-1}$, which is close to the reported value of 54.4 J mol$^{-1}$ K$^{-1}$ from free energy perturbation calculation (*Mukherjee and Schäfer, 2023*). However, the molar entropy is marginally reduced in all the macrostates of both αS and αS-fasudil systems, with values ranging between 55.01 J mol$^{-1}$ K$^{-1}$ to 55.25 J mol$^{-1}$ K$^{-1}$, with difference from that of bulk water being negligible. Upon closer investigation of the translational and rotational components of the total water entropy, we find that that the decrease of the translational component is more pronounced as compared to the rotational component. This implies that αS in its apo state or in the presence of fasudil has a comparatively greater impact on the translational component than on the rotational component of the entropy of water (see *Figure 8—figure supplement 1*). Nonetheless, the entropy analysis suggests that the presence of fasudil has an insignificant impact on the entropy of water.

Next, we estimated the protein conformational entropy using the PDB2ENTROPY program (*Fogolari et al., 2018*) that implements the Maximum Information Spanning Tree (MIST) approach (*King and Tidor, 2009*; *King et al., 2012*) on sets of torsional angles of the protein. The entropy is calculated from probability distributions of torsional degrees of freedom of the protein relative to random distribution i.e. corresponding to a fully flexible protein chain. The total protein entropy values for the metastable states in water (MS1 to MS3) and fasudil-bound states (FS1 to FS6) are shown in *Figure 8*. While the conformational entropies of the ligand-free states, MS1 to MS3, range from –72 to –75 J mol$^{-1}$ K$^{-1}$, the entropy of fasudil-bound states (FS1 to FS6) vary from –70 to –86 J mol$^{-1}$ K$^{-1}$, indicating that binding of fasudil to αS leads to a broadening of the span of entropy of these states. The highest entropy among the metastable states in the two environments correspond to FS5 and MS2 with values of –70.8 J mol$^{-1}$ K$^{-1}$ and –72.3 J mol$^{-1}$ K$^{-1}$, respectively. These states are also the most populated metastable states. MS1 and MS3 states have entropy values of –73.9 J mol$^{-1}$ K$^{-1}$ and –74.5 J mol$^{-1}$ K$^{-1}$, respectively. Notably, the presence of fasudil and its interactions with the protein chains can elicit the exploration of states that are more ordered in nature or increasingly deviated from a random chain. The largest deviation from a fully disordered chain is attained by the state FS1 with entropy value of –85.3 J mol$^{-1}$ K$^{-1}$, followed by FS2 having entropy of –83.1 J mol$^{-1}$ K$^{-1}$. FS4 is also relatively more ordered than FS5 with entropy of –76.2 J mol$^{-1}$ K$^{-1}$. FS3 and FS6 are closely disordered as FS5 with entropy values of –73.2 J mol$^{-1}$ K$^{-1}$ and –72.5 J mol$^{-1}$ K$^{-1}$, respectively. These observations of the total extent of disorderedness of the metastable states clearly indicate that the interactions of fasudil with αS residues can restrict their conformational degrees of freedom thus entrapping the protein in specific conformational states, thus manifesting as overall decrease in the protein entropy. Considering the diffuse binding of fasudil across several residues along the entire length of the protein, we further estimated and analysed how the entropic contributions are affected at the residue level (shown

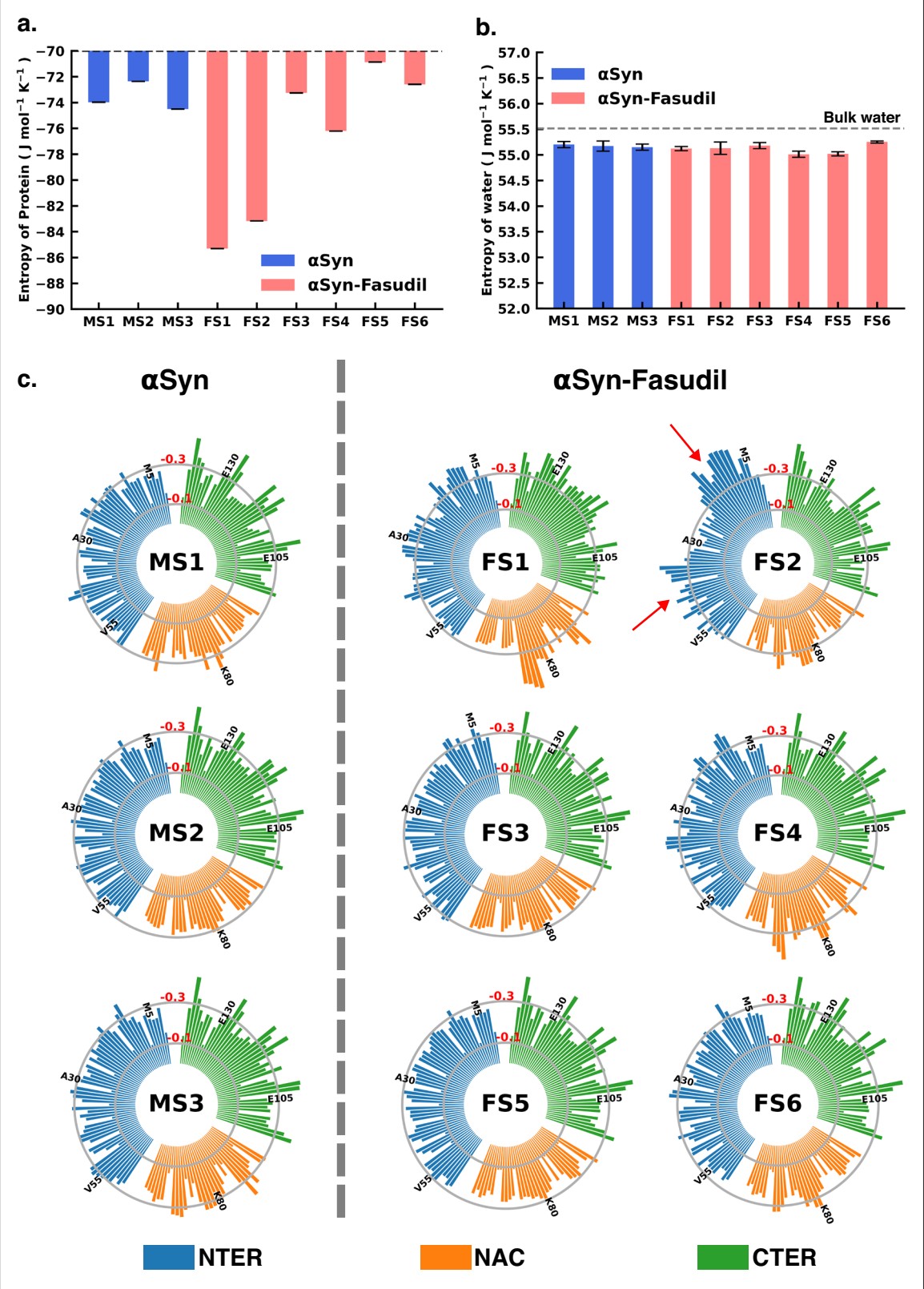

**Figure 8.** Entropy of αS system is modulated to a greater extent in presence of fasudil, with insignificant contribution from the solvent entropy. (**a**) Total Protein entropy of the metastable states of αS and αS-Fasudil calculated from the torsion angles, relative to a fully flexible chain (**b**) Total water entropy in αS and αS-Fasudil system and (**c**) Residue-wise backbone entropy within the αS and αS-Fasudil states.

*Figure 8 continued on next page*

*Figure 8 continued*

The online version of this article includes the following figure supplement(s) for figure 8:

**Figure supplement 1.** Translational and Rotational entropy of water.

**Figure supplement 2.** Residue-wise entropy of αS states populated in water and in the presence of fasudil.

**Figure supplement 3.** Residue-wise sidechain entropy of αS states populated in water and in the presence of fasudil.

in *Figure 8*, *Figure 8—figure supplements 2 and 3*). The total residue-wise entropy presented in *Figure 8—figure supplement 2* shows that the residues in the N-terminal region of fasudil-bound states FS1 and FS2, exhibit greater entropic fluctuations i.e. more negative values from the apo metastable states (MS1 to MS3), suggesting more ordered or restricted fluctuations of these residues. We note that the propensity of fasudil binding to the N-terminal residues is higher in FS1 and FS2, relative to other states (see *Figure 7*). We further decoupled the residue entropy into backbone and sidechain components; this analysis is depicted in *Figure 8*, *Figure 8—figure supplement 3*, respectively. Comparison of the sidechain entropies, computed from the $\chi$ torsion angles, for the unbound and bound states, does not show significant alteration upon small-molecule binding. Conversely, increased fluctuations in residue backbone entropies are observed in response to small-molecule binding (see *Figure 8*). The metastable states, MS1 to MS3, populated in water have minimally fluctuating residue entropies amongst the states. In contrast, significant fluctuations in residue entropies are noted predominantly in the N-terminal region (residues 1–60) and to some extent in the NAC region for states FS1 and FS2. These residues in the N-terminal region in FS1 and FS2 experience a drop in backbone entropy owing to the restricted degrees of freedom from fasudil binding. The decrease in residue backbone entropy correlates with fasudil binding hotspots and therefore can be ascribed to small-molecule interactions. Hydrophobic residues in the NAC region show decrease in entropy in ligand-binding locations mainly in state FS1, FS2, and FS4. However, intriguingly, we note that the correlation between fasudil interactions and residue entropy is not observed in state FS3. Finally, in the negatively charged C-terminal region, where fasudil preferentially targets the entire region in the αS protein, residue-wise entropic fluctuations are minimal compared to the apo state. Robustelli et al. refer to the interaction of fasudil with the C-terminal region as dynamic shuttling, in which breaking and formation of interactions with nearby residues occur simultaneously. This dynamic shuttling mechanism could possibly minimise the backbone entropic fluctuations compared to the relatively specific binding in the N-terminusand NAC regions.

## Exploring interplay of conformational entropy and mean first passage time in protein conformational transitions

The Mean First Passage Time (MFPT) for transition among the metastable states provides valuable information for understanding the dynamics and conformational changes in the system. MFPT rate defines the average time it takes for a system to transition between one metastable state to another. A lower MFPT value suggests a favourable transition among the states. The transition timescales for αS indicate that in the presence and absence of fasudil, the transition to the most populated state is preferred (see *Figure 9*). For αS-Fasudil system, the transition to the most populated states takes place over timescales ranging from 74 µs and 766.5 µs, whereas for the apo state it occurs between 29.8 µs and 37.4 µs. Upon comparing these transition times with those to other macrostates, it becomes evident that these timescales represent the shortest durations. This observation suggests that the transition to the MS2 and FS5 state in αS and αS-Fasudil system is the most favoured over all other states in the system.

While MFPT offers insights into the kinetics of the metastable states, the thermodynamic property, entropy, characterizes the diversity and disorder within the system. Although a state with higher population might not always have the higher entropy, we observed that in our case, the state with the highest population also has the highest protein entropy, for both αS and αS-Fasudil system. States with higher entropy are thermodynamically favored which in turn can affect the rates of state transitions, potentially influencing MFPT values. To decipher this, we have compared the transition times of each metastable state to the most populated state with the entropy of the state, and found a strong degree of correlation (see *Figure 9e*). This suggests that a state with higher entropy, which also has more number of accessible microstates, potentially provide multiple pathways for other macrostates

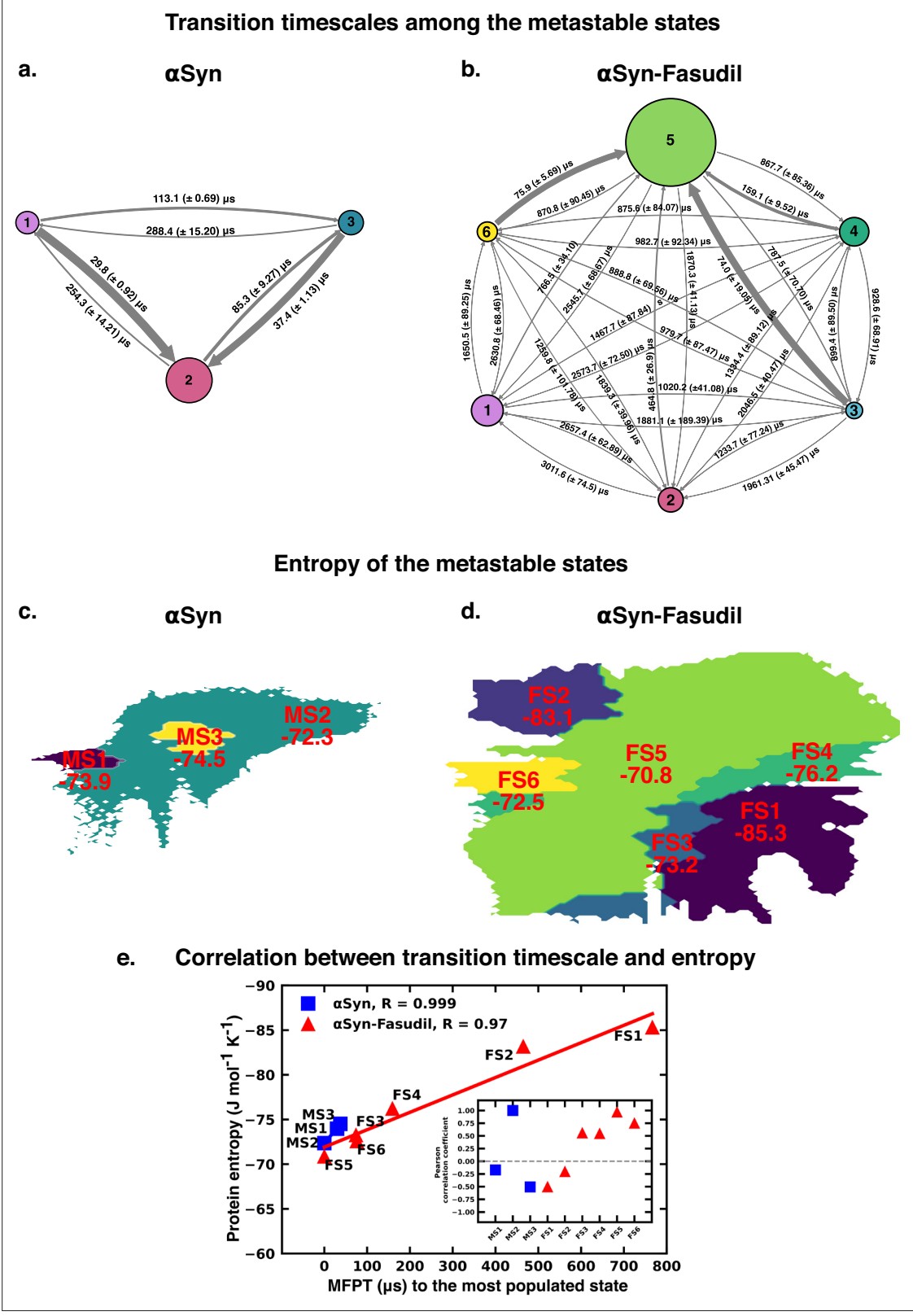

**Figure 9.** Mean first passage time for αS and αS-Fasudil ensembles indicates that a decrease in the transition time to the most entropic state in presence of fasudil, thereby potentially trapping S in the intermediate states. The rate network obtained from MFPT analysis for the (**a**) 3-state MSM of the αS ensemble in neat water and (**b**) 6-state MSM of the αS-Fasudil ensemble. The sizes of the discs is proportional to the stationary population of the respective state. The thickness of the arrows connecting the states is proportional to the transition rate (1/MFPT) between the two states and the MFPT

*Figure 9 continued*

values are shown on the arrows. (**c, d**) Projection of conformation sub-ensemble PCCA +for αS and αS-Fasudil system in latent space, with respective protein entropy values (in unit of J mol$^{-1}$ K$^{-1}$) annotated on top of it. (**e**) Correlation between protein entropy and transition time to the major state for αS and αS-Fasudil system. The inset plot corresponds to the correlation between entropy and transition to the macrostate for αS and αS-Fasudl system.

to access this state for both αS and αS-Fasudil system. However, as compared to αS in its apo state, the apparent increase (~2–20 times) in the time to transition to the major state in αS-Fasudil system, suggests that fasudil may potentially trap the protein conformations in the intermediate states, thereby slowing down αS in exploring the large conformational space in presence of fasudil.

## Conclusions

The conformationally dynamic disordered proteins challenge the application of conventional structure-based drug-design methods that relies on the specificity of a binding pocket, based on which the drug molecule is designed and optimized. A general framework describes the effect of small-molecule binding to IDPs by modulation of the disordered ensemble by increasing or decreasing its conformational entropy. Here, we characterised the effect of the small molecule, fasudil, on the conformational characteristics of the quintessential IDP, α-synuclein. Fasudil has been experimentally reported to curb the aggregation of αS both in vivo and in vitro. The ensemble view of long time scale atomic simulations of fasudil binding to αS monomer displays a fuzzy ensemble of αS with preferential binding of fasudil to the C-terminal region via charge-charge interactions and aromatic stacking. Using a deep neural-network based machine learning approach, variational autoencoder, we reconstructed a high dimensional feature that is Cα-pairwise distance into a two-dimensional latent space. We built an MSM using this data to delineate the metastable conformational states of αS and identify the differences that arise from small-molecule interactions. Comparison of the metastable states of αS in the presence and absence of the small molecule fasudil revealed that the small-molecule binding navigates the structural landscape of the protein. Importantly, more number of metastable states are populated indicating that small-molecule mediated conformational modulation led to entropy expansion (see *Figure 9d–e*). Atomic-level MD simulations have slowly emerged as a valuable tool for complementing experimental measurements of disordered proteins and providing detailed descriptions of their conformational ensembles (*Dedmon et al., 2005*; *Allison et al., 2009*; *Ullman et al., 2011*; *Esteban-Martín et al., 2013*). With improvements in force fields (*Best et al., 2014*; *Huang et al., 2017*; *Robustelli et al., 2018*) that model disordered proteins, the accuracy of MD simulations has dramatically increased, as assessed by their agreement with a wide variety of experimental measurements. Integrative approaches use a combination of experimental restraints such as chemical shifts, paramagnetic relaxation enhancement (PRE; *Dedmon et al., 2005*; *Bertoncini et al., 2005*; *Allison et al., 2009*; *Esteban-Martín et al., 2013*), residual dipolar coupling constants (RDCs; *Bertoncini et al., 2005*; *Bernadó et al., 2005*; *Marsh et al., 2008*), small-angle X-ray scattering (SAXS; *Bernadó et al., 2007*; *Ahmed et al., 2021*) and computational models from MD simulations to generate a more accurate description of the ensemble of disordered proteins. More recent advances include statistical approaches such as Bayesian inference models like Metainference (*Bonomi et al., 2016*), Experimental Inferential Structure Determination (EISD; *Brookes and Head-Gordon, 2016*; *Lincoff et al., 2020*) and Maximum Entropy formulations (*Roux and Weare, 2013*; *Boomsma et al., 2014*; *Cavalli et al., 2013*) that also take into account the experimental and back-calculation model errors and uncertainties. Such improved approaches to leverage MD simulations with improved force fields and experimental knowledge-based ensemble generation have shown tremendous potential for describing molecular recognition mechanisms of IDPs with other protein partners; scenarios such as folding-upon-binding (*Dyson and Wright, 2002*; *Robustelli et al., 2020*) or self-dimerisation. More importantly, a set of recent initiatives have focussed on molecular characterisation of the interaction of small-molecules with IDPs via effective combination of atomistic simulation and biophysical experiments (*Heller et al., 2020*; *Robustelli et al., 2022*).

In this regard, the present investigation takes a step ahead via a quantitative and statistical dissection of the αS ensemble in presence of a small-molecule. The implementation of Markov state model in the present work clearly brings out the feasibility of metastable states that would have a finite 'stationary' or equilibrium population in presence of small-molecule or in neat water. The uniqueness of these states is supported from the projection of their inter-residue Cα contact map of the macrostates

using DCVAE model. Our systematic investigation of the impact of fasudil on αS monomer ensemble reveals the altering effects of small-molecule interaction on αS ensemble by tuning the entropy of these states. Insights from studies targeting structural ensemble modulation by small-molecule can be effectively harnessed in designing drug-design strategies for aggregation diseases.

## Methods

### Monomer simulation protocol

A 1500 µs long all-atom MD simulation trajectory of αS monomer in aqueous fasudil solution was simulated by D. E. Shaw Research with the Anton supercomputerthat is specially purposed for running long-time-scale simulations (*Robustelli et al., 2022*). The protein was solvated in a cubic water-box with 108 sides and containing 50 mM concentration of Na or Cl ions. Protein, water molecules and ions present in the system were parameterised with the a99SB-*disp* force field (*Robustelli et al., 2018*). The small-molecule, fasudil, was parameterised using the generalised Amber force-field (GAFF). Production run of the system was performed in the NPT ensemble. Further details of the MD simulation can be obtained by referring to the original work by *Robustelli et al., 2022*. The extensive simulation trajectories were generously made available by D. E. Shaw Research upon request. To generate αS ensemble in water, we performed multiple simulations of αS monomer. An IDP is not represented by a fixed structure. Therefore, it would be most appropriate to run multiple simulations starting from different initial structures and simulate the local environment around those structures; thus generating

**Table 1.** The size of the simulation box for the 23 αS simulations.

| Conformation number | Cubic Box size (nm) |
| --- | --- |
| 1 | 12.14 |
| 2 | 11.33 |
| 3 | 13.55 |
| 4 | 10.41 |
| 5 | 10.97 |
| 6 | 10.44 |
| 7 | 10.10 |
| 8 | 11.44 |
| 9 | 10.57 |
| 10 | 10.75 |
| 11 | 11.34 |
| 12 | 10.35 |
| 13 | 13.38 |
| 14 | 10.08 |
| 15 | 10.38 |
| 16 | 11.78 |
| 17 | 11.92 |
| 18 | 9.37 |
| 19 | 10.15 |
| 20 | 11.16 |
| 21 | 11.24 |
| 22 | 11.46 |
| 23 | 12.00 |

an ensemble effectively sampling the phase space. Accordingly, for initiating the apo simulations, instead of biasing the initial structure (using the starting structure used for simulations with fasudil), we randomly chose 23 different conformations from previously reported (*Robustelli et al., 2018*) multi-μs αS trajectory in water. We have also determined the $R_g$ values corresponding to the starting structures used to start the apo simulations. It is evident that the 23 starting conformations chosen represent the entire span of the $R_g$ space that is sampled in the fasudil ensemble (see *Figure 1—figure supplement 1*). In addition, we have also calculated the RMSD of the starting structures with respect to the starting structure of the fasudil simulation (see *Figure 1—figure supplement 7*), secondary structure content (see *Figure 1—figure supplements 8–10*) and solvent accessible surface area (SASA; see *Figure 1—figure supplements 11–12*) to validate our choice of starting structures.The simulation details are described below. Each protein conformation is solvated in a cubic box (see *Table 1*) of explicit water molecules and Na or Cl ions were added to a concentration of 50 mM, to maintain the same conditions as used in the simulations with fasudil. The all-atom a99SB-*disp* force field and water model (*Robustelli et al., 2018*) were used to simulate the protein and solvent. The unbiased MD simulations were performed using the Gromacs simulation package (*Lindahl et al., 2020*). A timestep of 2 fs and the leap-frog integrator was used. The simulation was performed in the isothermal-isobaric (NPT) ensemble at an average temperature of 300 K, maintained using the v-rescale thermostat (*Bussi et al., 2007*) with a relaxation time of 0.1 ps and a pressure of 1 bar using the Parrinello-Rahman barostat (*Parrinello and Rahman, 1981*) with a time constant of 2 ps for coupling. The Verlet cutoff scheme (*Páll and Hess, 2013*) with 1.0 nm cutoff was applied for Lennard-Jones interactions and short-range electrostatic interactions. Long-range electrostatic interactions were computed using the Particle-Mesh Ewald (PME) summation method (*York et al., 1993*) and covalent bonds involving hydrogen atoms were constrained using the LINCS algorithm (*Hess et al., 1997*). All the systems were minimized using the steepest descent algorithm, followed by equilibration in the isothermal-isochoric (NVT) and subsequently in the NPT ensemble with position restraints on the heavy atoms of the protein. Twenty-three simulations, each starting from different conformations, were performed. These simulation timescales are variable, ranging from 1 to 4 μs, amounting to a cumulative length of ~62 μs.

We have compared the ensemble properties of the αS in the presence and absence of fasudil. As seen in *Figure 1—figure supplement 1*, the distributions of $R_g$ for both the simulations have a significant overlap, thereby indicating that in terms of extent of compaction, the conformational sampling of the protein is comparable in the presence and absence of fasudil. We have also compared other structural features such as end-to-end distance ($R_{ee}$; see *Figure 1—figure supplement 2*), SASA (see *Figure 1—figure supplement 3*) and secondary structure content (see *Figure 1—figure supplements 4–6*). We have found that these features are highly comparable for both the ensembles, which clearly indicates that the αS simulations we performed from different starting structures have effectively sampled the phase space as the single long simulation of the αS-Fasudil system.

## Dimension reduction using β-variational autoencoder

The probability distribution $p(x)$ of a multi-dimensional variable $x$ can be determined using various approaches (*Bengio and Bengio, 2000*; *Larochelle and Murray, 2011*), one of them being the latent variable model. In this model, the probability distribution is modeled as a joint distribution $p(x, z)$ of the observable variable $x$ and the hidden latent variable $z$ which is given as $p(x, z) = p(x|z)p(z)$, where $p(z)$ is the prior distribution over the latent variables $z$ and $p(x|z)$ is the likelihood of generating a true data, given $z$. The $p(x)$ is determined by marginalization over all the latent variables giving us,

$$p(x) = \int p(x|z)p(z)dz \tag{2}$$

However, this integral is often intractable thereby constraining the calculation of the true posterior distribution $p(z|x)$. This can be solved using, variational inference, which aims to estimate the true posterior using an approximate distribution $q(z|x)$, using the encoder part of VAE which parameterizes the distribution using a mean vector and a variance vector. The goal is to determine $q(z|x)$ which is closest to the true posterior by minimizing the Kullback-Leibler (KL) divergence between them which is given as

$$D_{KL}(q(z|x)\|p(z|x)) = \int q(z|x) \log\left(\frac{q(z|x)}{p(z|x)}\right) dz \tag{3}$$

Upon expanding and rearranging, we have

$$\log p(x) - D_{KL}(q(z|x)\|p(z|x)) = \mathbb{E}_{z \sim q(z|x)} \log p(x|z) - D_{KL}(q(z|x)\|p(z)) \tag{4}$$

where $p(x)$ is the evidence. The VAE is trained so that we maximize the log-likelihood of the evidence and minimize the KL divergence between the estimated and true posterior. The loss function of VAE is given by the negation of *Equation 2* and is given as,

$$\text{Loss}_{VAE} = -\mathbb{E}_{z \sim q(z|x)} \log p(x|z) + D_{KL}(q(z|x)\|p(z)) \tag{5}$$

The training objective of VAE being minimizing this loss function to generate a realistic data. The VAE model can be further improved by keeping the distance between the approximated and the true posterior under a small distance, $\delta$. This is achieved by maximizing the equation

$$\mathcal{F}(\beta; x, z) = \mathbb{E}_{z \sim q(z|x)} \log p(x|z) - \beta(D_{KL}(q(z|x)\|p(z)) - \delta) \tag{6}$$

which modifies the loss function to

$$\text{Loss}_{\beta-VAE} = -\mathbb{E}_{z \sim q(z|x)} \log p(x|z) + \beta D_{KL}(q(z|x)\|p(z)) \tag{7}$$

where β is the Lagrangian multiplier (under the Karush-Kuhn-Tucker conditions). This enhances the probability of generating an actual data and is known as β-VAE (*Higgins et al., 2016*). The expectation term in the loss function of β-VAE requires a point $z$ to be sampled from the posterior distribution. This is performed using the reparametrization trick

$$z = \mu + \sigma \odot \epsilon \tag{8}$$

where $\epsilon$ is sampled from a Standard Normal distribution. This makes the VAE model trainable using the backpropagation algorithm.

In our work we have used $C_\alpha$-$C_\alpha$ pairwise distance (excluding $i$, $i+1$ and $i+2$ distances) sampled at every 9 ns of the entire 1.5 millisecond of the αS protein in presence of fasudil trajectory as our input. We have built our model using Tensorflow backend (*Abadi et al., 2016*) with Keras library. Training was performed with 80% of the data while 20% was used for testing. We have employed a symmetric VAE where the input layer consists of 9453 nodes followed by four hidden layers having 4096, 512, 128 and 16 nodes respectively, and 2 nodes in the latent layer. Hyperbolic tangent (*tanh*) is used as our activation function with weights being initialized using the *glorot_uniform* (*Glorot and Bengio, 2010*) method in all the layers except for the output layer where we have used *sigmoid* activation. Adaptive Moment Estimation (Adam *Kingma and Ba, 2014*) was used as the optimizer with a learning rate of $1 \times 10^{-4}$. The model was trained for 300 epochs with a batch size of 64. The relevant Python implementations are provided in Github (copy archived at *JMLab-tifrh, 2024*).

## Building the denoising convolutional variational autoEncoder

The Denoising Convolutional Variational Autoencoder (DCVAE) model employs five convolutional layers which incorporate filters of sizes 16, 32, 64, 96, and 128, with corresponding kernels of size 3x3. Strides are set at 1, 2, 2, 2, 2, and 1 for efficient feature extraction. This is followed by six densely connected layers with neuron counts of 4096, 2048, 1024, 256, 64 and 16 in the successive dense layers. The latent layer is comprised of 2 neurons. The decoder mirrors the encoder architecture, to reconstruct back the input contact maps. Hyperbolic tangent activation is used in all the layers, excluding the output layer where *sigmoid* activation is used. Adam optimizer with a learning rate of $5 \times 10^{-4}$ is used for training the DCVAE. The model was trained for 300 epochs with a batch size of 32 with $\beta = 1 \times 10^{-12}$. Training was performed with 90% of the data while 10% was used for testing.

At first, we added a noise to 200 replicas of contact map for each of the states of αS and αS-fasudil. The noise was generated stochastically from a standard normal distribution which was multiplied by a noise factor of 0.035 and added to each of the replicas, followed by scaling the dataset so that the values range between 0 and 1 using the formula

$$x_{scaled} = \frac{x - x_{min}}{x_{max} - x_{min}} \tag{9}$$

where $x_{min}$ and $x_{max}$ are the minimum and maximum values of the input contact map. In order to assess the denoising performance, we have calculated the structural similarity index measure (SSIM) values (*Hore and Ziou, 2010*). The SSIM value can range between –1 to +1, thereby indicating the structural dissimilarity or similarity between the original ($x$) and denoised ($y$) contact maps. Mathematically, this is expressed as

$$\text{SSIM}(x, y) = [l(x, y)]^\alpha \cdot [c(x, y)]^\beta \cdot [s(x, y)]^\gamma \tag{10}$$

where the functions $l(x, y)$, $c(x, y)$, and $s(x, y)$ compare the luminance, contrast and structure between the contact maps and is given as

$$\begin{aligned} (x, y) &= \frac{2\mu_x\mu_y + C_1}{\mu_x^2 + \mu_y^2 + C_1} \\ c(x, y) &= \frac{2\sigma_x\sigma_y + C_2}{\sigma_x^2 + \sigma_y^2 + C_2} \\ s(x, y) &= \frac{\sigma_{xy} + C_3}{\sigma_x\sigma_y + C_3} \end{aligned} \tag{11}$$

The variables $\mu_i$ and $\sigma_i$ represent the pixel sample mean and variance of $i$, where $i = \{x, y\}$. $\sigma_{xy}$ is the covariance of the $x$ and $y$ contact maps. The exponents $\alpha$, $\beta$ and $\gamma$ controls the weighted contribution to the SSIM value and are set to 1. The constants $C_1 = (k_1 L)^2$, $C_2 = (k_2 L)^2$ are small positive values introduced to prevent instability and division by zero. They are used for numerical stability and to avoid problems when the means and variances of the contact maps are close to zero. We have set $k_1 = 0.01$, $k_2 = 0.03$, and $C_3 = C_2/2$ as per convention (*Wang et al., 2004*).

In addition, we have also calculated the peak Signal-to-Noise Ratio to evaluate our denoising performance and is calculated as

$$\text{PSNR} = 10 \cdot \log_{10}\left(\frac{R^2}{\text{MSE}}\right) \tag{12}$$

where, $R$ is the maximum pixel value (255 in our case) and MSE is the mean-square error between the original and denoised image and is given as

$$\text{MSE} = \frac{1}{MN}\sum_{i=1}^{M}\sum_{j=1}^{N}[\mathcal{O}_{ij} - \mathcal{D}_{ij}] \tag{13}$$

where M is the height (number of rows) of the image, N is the width (number of columns) of the image and $\mathcal{O}_{ij}$ and $\mathcal{D}_{ij}$ is the pixel intensity at position ($i$, $j$) for the original and denoised image, respectively. The relevant Python implementations are provided in Github (copy archived at *JMLab-tifrh, 2024*).

## Building a Markov State Model of monomer ensemble

The 2D latent space reconstructed using β-VAE described in the previous subsection was then used to build an MSM to elicit the metastable states underlying the ensembles. We employed PyEMMA (*Scherer et al., 2015*), an MSM building and analysis package. The 2D data is subjected to $k$-means clustering. We used the VAMP-2 score to determine the number of microstates. By varying the number of microstates from 10 to 220, we found that the VAMP-2 score saturated at higher numbers for both αS and αS-Fasudil simulations. We find that the VAMP-2 score has saturated at 180 microstates (see *Figure 3—figure supplements 10–11*). A transition matrix was then built by counting the number of transitions among the microstates at a specific lag time. To choose the lag time, the implied timescale (ITS) or relaxation timescales of the systems were calculated over a range of lag times and plotted as a function of lag time. The timescale at which the ITS plot levels off is chosen as the lag time to build the final MSM. The ITS plots corresponding to the neat water and aqueous fasudil systems are presented in *Figure 3*. Accordingly, MSM lag times of 36 and 32 ns were selected for neat water fasudil systems, respectively. At these lag times, by identifying the gaps between the ITS, a three-state model for neat water and a six-state model for aqueous fasudil system was built. Lastly, the transition path theory (*Weinan and Vanden-Eijnden, 2006*; *Metzner et al., 2009*; *Noé et al., 2009*) was used to ascertain the transition paths, fluxes and timescales in these models. Bootstrapping was performed to estimate the errors associated with the state populations and transition timescales. In ten iterations, the model

was rebuilt after eliminating a randomly selected trajectory followed by calculation of the state populations and transition timescales. The mean value and standard deviations of the values collected by bootstrapping are reported. A Chapman-Kolmogorov test was performed for the MSMs of the neat water and small molecule ensembles, that tests the validity of the model prediction at longer time scales (see *Figure 3—figure supplement 3*).

## Entropy of water

In order to calculate the entropy of water, simulations were performed using GROMACS (version 2022.3). A single snapshot was chosen from each macrostate of α-synuclein in its apo state and in presence of fasudil, which were then solvated in a cubic box, placed within 1 nm from the box edge. The system was neutralized using NaCl and the salt concentration was kept at 50 mM. The system was energy minimized using the steepest descent algorithm, followed by equilibration in the canonical ensemble for 2 ps. The average temperature was maintained at 300 K using velocity rescale thermostat with a time constant of 0.1 ps. Following this, NPT equilibration was performed for 2 ps to maintain the pressure at 1.0 bar using the Berendsen barostat. Finally, three independent production runs were performed from the equilibrated structures in the NPT ensemble using the velocity-verlet integrator with a timestep of 1 fs for a timescale of 20 ps with frames saved at every 4 fs. In addition, we have performed three simulations of neat water using the parameters as described for the macrostates.

The entropy of water was estimated using the DoSPT program (*Caro et al., 2016*; *Caro et al., 2017*) that computes the thermodynamic properties from MD simulation in the framework of the 2-phase-thermodynamics (2PT) method (*Lin et al., 2003*; *Lin et al., 2010*). This is achieved from calculating the density of state (DoS) function, $I(v)$ which is the Fourier transform of the velocity auto correlation function,

$$I(v) = \frac{1}{k_B T} \sum_{l=1}^{N} \sum_{k=1}^{3} \lim_{\tau \to \infty} \frac{m_l}{\tau} \left| \int_{-\tau}^{\tau} v_l^k(t) \exp(-2i\pi vt) \, dt \right|^2 \tag{14}$$

where $m_l$ is the mass of atom $l$, $v_l^k$ is the velocity of atom $l$ along the $k$ direction and $N$ is the total number of atoms in the system. DoS can be represented as a sum of translational (trn), rotational (rot), and vibrational (vib) motions (*Lin et al., 2010*).

$$I(v) = I_{trn}(v) + I_{rot}(v) + I_{vib} \tag{15}$$

For water, the 2PT model overcomes the anharmonic nature of the low frequency modes (diffusive motions and libration) by decomposing the DoS into solid-like ($I^s$) and gas-like ($I^g$) contribution for the translational and rotational components. The solid-like DoS is treated using the harmonic oscillator model, whereas the gas-like DoS is described using the Enskog hard sphere (HS) theory for translation and the rigid rotor (RR) model for rotation. The entropy is calculated by integrating the DoS, weighted by the corresponding weighting functions $W$,

$$S_k = k_B \left[ \int_0^\infty I_k^s(v) W_k^s(v) dv + \int_0^\infty I_k^g(v) W_k^g dv \right] \tag{16}$$

where $k$ = trn, rot, or vib and the weighting functions are given as

$$\begin{aligned} W^s(v) &= \frac{\beta h v}{\exp(\beta h v) - 1} - \ln[1 - \exp(-\beta h v)] \\ W_{trn}^g &= \frac{S^{HS}}{3k_B} \\ W_{rot}^g &= \frac{S^{RR}}{3k_B} \end{aligned} \tag{17}$$

where $\beta = (k_B T)^{-1}$ and $h$ is the planck constant (*Huang et al., 2011*).

## Protein conformational entropy

We calculated the protein conformational entropy using the program PDB2ENTROPY (*Fogolari et al., 2018*). This program is based on the nearest-neighbor approach from probability distributions

of torsion angles at a given temperature relative to uniform distributions. The entropy is computed using the Maximum Information Spanning Tree (MIST) approach (*King and Tidor, 2009*; *King et al., 2012*).

## Acknowledgements

We sincerely thank D E Shaw research for providing us with access to long simulation trajectories of monomeric alpha-synuclein in presence of fasudil which seeded the project. We acknowledge support of the Department of Atomic Energy, Government of India, under Project Identification No. RTI 4007. JM acknowledges Core Research grants provided by the Department of Science and Technology (DST) of India (CRG/2023/001426).

## Additional information

### Funding

| Funder | Grant reference number | Author |
|---|---|---|
| Department of Atomic Energy, Government of India | RTI 4007 | Jagannath Mondal |
| Department of Science and Technology, Ministry of Science and Technology, India | CRG/2023/001426 | Jagannath Mondal |

The funders had no role in study design, data collection and interpretation, or the decision to submit the work for publication.

### Author contributions

Sneha Menon, Data curation, Formal analysis, Validation, Investigation, Visualization, Methodology, Writing – original draft, Writing – review and editing; Subinoy Adhikari, Data curation, Software, Formal analysis, Validation, Investigation, Visualization, Methodology, Writing – original draft, Writing – review and editing; Jagannath Mondal, Supervision, Funding acquisition, Validation, Investigation, Writing – original draft, Project administration, Writing – review and editing

### Author ORCIDs

Sneha Menon  http://orcid.org/0000-0003-1079-8928
Jagannath Mondal  https://orcid.org/0000-0003-1090-5199

Reviewer #2 (Public Review): https://doi.org/10.7554/eLife.97709.3.sa1
Reviewer #3 (Public Review): https://doi.org/10.7554/eLife.97709.3.sa2
Author response https://doi.org/10.7554/eLife.97709.3.sa3

## Additional files

### Supplementary files
• MDAR checklist

### Data availability

The raw files of the data presented in this article and the key simulation trajectories have been uploaded to zenodo (https://doi.org/10.5281/zenodo.14177307) for public access. The implementations of Machine learning protocols are publicly available in our group GitHub webpage (https://github.com/JMLab-tifrh/Protein_Ligand_Variational_Autoencoder; copy archived at *JMLab-tifrh, 2024*).

The following dataset was generated:

| Author(s) | Year | Dataset title | Dataset URL | Database and Identifier |
|---|---|---|---|---|
| Subinoy A, Sneha M, Jagannath M | 2024 | An Integrated Machine Learning Approach Delineates an Entropic Expansion Mechanism for the Binding of a Small Molecule to α-Synuclein | https://doi.org/10.5281/zenodo.14177307 | Zenodo, 10.5281/zenodo.14177307 |

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
