## [Editor Report · eLife Assessment]

This study describes the application of machine learning and Markov state models to characterize the binding mechanism of alpha-Synuclein to the small molecule Fasudil. The results suggest that entropic expansion can explain such binding. However, the simulations and analyses in their present form are **inadequate**.

---

## [Referee Report · Reviewer #2 (Public Review)]

The manuscript by Menon et al describes a set of simulations of alpha-Synuclein (aSYN) and analyses of these and previous simulations in the presence of a small molecule.

Comments on latest version:

I have read the authors' response to my comments as well as to the other reviewers. Summarizing briefly, I don't think they provide substantial answer to the questions/comments by me or reviewer 3, and generally do not quantify the results/effects data. I still remain unconvinced about the analyses and conclusions. Rather than rewriting another set of comments, I think it will be more useful for all (authors and readers) simply to be able to see the entire set of reviews and responses together with the paper.

---

## [Referee Report · Reviewer #3 (Public Review)]

In this manuscript Menon, Adhikari, and Mondal analyze explicit solvent molecular dynamics (MD) computer simulations of the intrinsically disordered protein (IDP) alpha-synuclein in the presence and absence of a small molecule ligand, Fasudil, previously demonstrated to bind alpha-synuclein by NMR spectroscopy without inducing folding into more ordered structures. In order to provide insight into the binding mechanism of Fasudil the authors analyze an unbiased 1500us MD simulation of alpha-synuclein in the presence of Fasudil previously reported by Robustelli et.al. (Journal of the American Chemical Society, 144(6), pp.2501-2510). The authors compare this simulation to a very different set of apo simulations: 23 separate1-4us simulations of alpha-synuclein seeded from different apo conformations taken from another previously reported by Robustelli et. al. (PNAS, 115 (21), E4758-E4766), for a total of ~62us.

To analyze the conformational space of alpha-synuclein - the authors employ a variational auto-encoder (VAE) to reduce the dimensionality of Ca-Ca pairwise distances to 2 dimensions, and use the latent space projection of the VAE to build Markov state Models. The authors utilize k-means clustering to cluster the sampled states of alpha-synuclein in each condition into 180 microstates on the VAE latent space. They then coarse grain these 180 microstates into a 3-macrostate model for apo alpha-synuclein and a 6-macrostate model for alpha-synuclein in the presence of fasudil using the PCCA+ course graining method. Few details are provided to explain the hyperparameters used for PCCA+ coarse graining and the rationale for selecting the final number of macrostates.

The authors analyze the properties of each of the alpha-synuclein macrostates from their final MSMs - examining intramolecular contacts, secondary structure propensities, and in the case of alpha-synuclein:Fasudil holo simulations - the contact probabilities between Fasudil and alpha-synuclein residues.

The authors utilize an additional variational autoencoder (a denoising convolutional VAE) to compare denoised contact maps of each macrostate, and project onto an additional latent space. The authors conclude that their apo and holo simulations are sampling distinct regions of the conformational space of alpha-synuclein projected on the denoising convolutional VAE latent space.

Finally, the authors calculate water entropy and protein conformational entropy for each microstate. To facilitate water entropy calculations - the author's take a single structure from each macrostate - and ran a 20ps simulation at a finer timestep (4 femtoseconds) using a previously published method (DoSPT), which computes thermodynamic properties of water from MD simulations using autocorrelation functions of water velocities. The authors report that water entropy calculated from these individual 20ps simulations is very similar.

For each macrostate the authors compute protein conformational entropy using a previously published Maximum Information Spanning tree approach based on torsion angle distributions - and observe that the estimated protein conformational entropy is substantially more negative for the macrostates of the holo ensemble.

The authors calculate mean first passage times from their Markov state models and report a strong correlation between the protein conformational entropy of each state and the mean first passage time from each state to the highest populated state.

As the authors observe the conformational entropy estimated from macrostates of the holo alpha-synuclein:Fasudil is greater than those estimated from macrostates of the apo holo alpha-synuclein macrostates - they suggest that the driving force of Fasudil binding is an increase in the conformational entropy of alpha-synuclein. No consideration/quantification of the enthalpy of alpha-synuclein Fasudil binding is presented.

Strengths:

The author's utilize MD simulations run with an appropriate force field for IDPs a99SB-disp and a99SB-disp water (Robustelli et. al, PNAS, 115 (21), E4758-E4766) - which has previously been used to perform MD simulations of alpha-synuclein that have been validated with extensive NMR data.

The contact probability between Fasudil and each alpha-synuclein residue observed in the previously performed 1500us MD simulation of alpha-synuclein in the presence of Fasudil (Robustelli et. al., Journal of the American Chemical Society, 144(6), pp.2501-2510) was previously found to be in good agreement with experimental NMR chemical shift perturbations upon Fasudil binding - suggesting that this simulation is a reasonable choice for understanding IDP:small molecule interactions.

Comments on the latest version:

While the authors have provided additional information in the updated manuscript, none of the additional analyses address the fundamental flaws of the manuscript.

The additional analyses do not convincingly demonstrate that these two extremely different simulation datasets (1500 microsecond unbiased MD for a-synuclein + fasudil, 23 separate 1-4 microsecond simulations of apo a-synuclein) are directly comparable for the purposes of building MSMs.

The additional analyses do not demonstrate that there are sufficient conformational transitions among kinetically metastable states observed in 23 separate 1-4 microsecond simulations of apo a-synuclein to build a valid MSM, or that the latent space of the VAE is kinetically meaningful.

If one is interested in modeling the kinetics and thermodynamics of transitions between a set of conformational states, and they run a small number of MD simulations that are too short to see conformational transitions between conformational states - any kinetics and thermodynamics modeled by an MSM will be inherently meaningless. This is likely to be the case with the apo a-synuclein dataset analyzed in this investigation.

Simulations of 1-4 microseconds are almost certainly far too short to see a meaningful sampling of conformational transitions of a highly entangled 140-residue IDP beyond a very local relaxation of the starting structures, and the authors provide no analyses to suggest otherwise.

Without convincingly demonstrating reasonable statistics of conformational changes from the very small apo simulation dataset analyzed here, it seems highly likely the apparent validity of the apo MSM results from learning a VAE latent space that groups structurally and kinetically distinct conformations into similar states, creating the spurious appearance of transitions between states. As such, the kinetics and thermodynamics of the resulting MSM are likely to be relatively meaningless, and comparisons with an MSM for a-synuclein in the presence of fasudil are likely to be meaningless.

In its present form, this study provides an example of how the use of black-box machine learning methods to analyze molecular simulations can lead to obtaining misleading results (such as the appearance of a valid MSM) - when more basic analyses are omitted.

---

## [Author Response]

The following is the authors’ response to the current reviews.

**Public Reviews:**

**Reviewer #2 (Public Review):**
I have read the authors' response to my comments as well as to the other reviewers. Summarizing briefly, I don't think they provide substantial answer to the questions/comments by me or reviewer 3, and generally do not quantify the results/effects data. I still remain unconvinced about the analyses and conclusions. Rather than rewriting another set of comments, I think it will be more useful for all (authors and readers) simply to be able to see the entire set of reviews and responses together with the paper.

The authors disagree with the views of referees. The authors have provided point-wise precise responses to each of the previous comments. The authors find that the referee has not been able to engage with the responses and accompanying analysis that were provided while communicating the previous response.

The following extensive analyses were performed by the authors while submitting our revision of round 2 of peer-review to address the comments of reviewer 2 and reviewer 3 that were raised by them on the previous versions:

(1) We calculated the distribution of multiple metrics for both the apo and holo simulations, including their secondary structure composition, and demonstrated the robustness of our findings.

(2) We analyzed smaller 60 µs chunks from two parts of the 1.5 ms trajectory and showed how, in combination with the Markov state modeling (MSM) approach, these chunks effectively capture equilibrium properties.

(3) We thoroughly investigated the choice of starting structures, examining parameters such as Rg, RMSD, secondary structure, and SASA, in response to Referee 3's concerns about the objectivity of our dimension reduction approach.

(4) We conducted multiple analyses using VAMP-scores and justified the use of a Variational Autoencoder (VAE) over tICA.

(5) We had extensively verified the choice of hyperparameters used in constructing the MSM.

(6) To aleviate referee concerns, we had retrained a VAE with four latent dimensions and used it to build an MSM, ensuring the robustness of our approach.

However, we find that Referee has not considered these additional analysis in response to his/her comments on the manuscript.

Since referee 2 also draws comments from Referee 3, it is worth noting that some of the comments from Referee 2 and Referee 3 in Round 1 were mutually contradictory. In particular, Referee 3's suggestion in Round 1 to use the same initial configuration for simulations of intrinsically disordered proteins (IDPs) in both apo and ligand-bound forms contradicts the fundamental principle that IDPs should not possess structural bias. This recommendation also directly conflicts with Referee 2's request for greater diversity in starting structures. Our manuscript provided robust evidence that our initial configurations are indeed diverse, with one configuration coincidentally matching that used in the ligand-bound simulations. Despite this, we addressed both sets of concerns in our Round 2 revisions. Unfortunately, it seems that these efforts were overlooked in the subsequent round of review.

Referee 2's suggestion in prevous round of review comments to mix both holo and apo simulation trajectories for MSM construction is conceptually wrong and indicates a lack of understanding of transition matrix building in this field. Nevertheless, we addressed these comments by performing additional analyses and demonstrating the robustness of our current MSM.

**Reviewer #3 (Public Review):**
Summary:While the authors have provided additional information in the updated manuscript, none of the additional analyses address the fundamental flaws of the manuscript.The additional analyses do not convincingly demonstrate that these two extremely different simulation datasets (1500 microsecond unbiased MD for a-synuclein + fasudil, 23 separate 1-4 microsecond simulations of apo a-synuclein) are directly comparable for the purposes of building MSMs.

The 23 unbiased 1-4 microsecond simulations of apo αS totals to ~ 60 us.

**Author response image 1. sa3fig1:** Left figure : Distribution of the radius of gyration (Rg) of the 23 apo simulation (as shown in the colourbar) and holo simulation (black). Right figure : Mean and standard deviation (as error bar) of the Rg of the 23 apo (colourbar) and holo simulations (black).

We have plotted the distribution of the Radius of gyration (Rg) for the 23 apo simulation (colour bar) and the holo simulation (black) as shown in the left figure and also compared the mean and standard deviations of the Rg values (right figure). We find that our apo simulations span the entire space of Rg as is spanned by the holo simulation. We have also measured the mean and standard deviations (SD) (horizontal error bar) of the apo and holo simulations. The fact that the apo simulations have mean and SDs comparable to those of the holo ensemble suggests that the majority of the apo simulations are sampling similar conformational space as those observed in the ligand-bound holo form and hence can be used for building the MSM.

The additional analyses do not demonstrate that there are sufficient conformational transitions among kinetically metastable states observed in 23 separate 1-4 microsecond simulations of apo a-synuclein to build a valid MSM, or that the latent space of the VAE is kinetically meaningful.

We have performed the Chapman-Kolmogorov test to compare observed and predicted transition probabilities over increasing lag times and found good agreement between these probabilities, thereby suggesting that transitions between states are well-sampled for both the apo (Author response image 2) and holo simulation (Figure S9).

**Author response image 2. sa3fig2:** The Chapman-Kolmogorov test performed for the three state Markov State Model of the αS ensemble.

As for the latent space of VAE, we have compared the VAMP2 score and compared with tICA. VAE has a higher VAMP2 score as compared to tICA thereby indicating its efficacy in capturing slower mode for both apo and holo simulation (Fig. S7 and S8).

If one is interested in modeling the kinetics and thermodynamics of transitions between a set of conformational states, and they run a small number of MD simulations that are too short to see conformational transitions between conformational states - any kinetics and thermodynamics modeled by an MSM will be inherently meaningless. This is likely to be the case with the apo asynuclein dataset analyzed in this investigation.

We disagree with the referee’s view. The referee does not seem to understand the point of building Markov state models via short-time scale trajectories. The distribution of Rg of all the 23 apo simulations spans the entire Rg space sampled by the holo simulation, thereby suggesting that multiple short simulations can sample structures of varying sizes as sampled from the 1.5 ms holo simulation (see Author response image 1).

Simulations of 1-4 microseconds are almost certainly far too short to see a meaningful sampling of conformational transitions of a highly entangled 140-residue IDP beyond a very local relaxation of the starting structures, and the authors provide no analyses to suggest otherwise.

**Author response image 3. sa3fig3:** Autocorrelation of the first principal component of the backbone dihedral for the apo (colourbar) and holo (black) simulation.

**Author response image 4. sa3fig4:** Autocorrelation of the second principal component of the backbone dihedral for the apo (colourbar) and holo (black) simulation.

In order to assess the 23 short simulations in capturing meaningful kinetics and thermodynamics, we have computed the backbone dihedrals which were then reduced to two principal components for both the 23 apo and holo simulations. We then calculated the autocorrelation time for each of the components and for each of the apo and holo simulations which are plotted in Author response image 3 and Author response image 4 respectively.

The autocorrelation for the holo and most of the apo simulation is similar, thereby suggesting that there is sufficient sampling of conformational transitions between conformational states in the apo simulations and are therefore able to represent the structural changes of the system similarly to the long simulation.

Without convincingly demonstrating reasonable statistics of conformational changes from the very small apo simulation dataset analyzed here, it seems highly likely the apparent validity of the apo MSM results from learning a VAE latent space that groups structurally and kinetically distinct conformations into similar states, creating the spurious appearance of transitions between states. As such, the kinetics and thermodynamics of the resulting MSM are likely to be relatively meaningless, and comparisons with an MSM for a-synuclein in the presence of fasudil are likely to be meaningless.

We have shown above that the short simulations are able to capture the structural changes in the long simulation. In addition we have compared the VAMP2 score of the apo and holo simulation with tICA and found out that VAE is superior in capturing long timescale dynamics, for both apo and holo simulation (Fig. S7 and S8).

In its present form, this study provides an example of how the use of black-box machine learning methods to analyze molecular simulations can lead to obtaining misleading results (such as the appearance of a valid MSM) - when more basic analyses are omitted.

The authors disagree with the referee’s viewpoint on our manuscript. We find that the majority of the contents of the referee’s comments are cursory and lack objectivity.

The referee’s loose reference on Machine learning as a *black box* lacks basic knowledge to comprehend artificial deep neutral network’s long-proven ability to objectively deduce optimal set of lower-dimensional representation of conformational subspace of complex biomacromolecule. The referee’s views on the manuscript ignore the extensive optimization of hyper-parameters that were carried out by the authors in developing the suitable framework of beta-variational autoencoder for deducing optimal latent space representation of complex and fuzzy conformational landscape of an IDP such as alpha-synuclein. We had thoroughly investigated the choice of starting structures, examining parameters such as Rg, RMSD, secondary structure, and SASA, in response to Referee 3's concerns about the objectivity of our dimension reduction approach. However, we find that referee 3 has ignored the analysis provided to justify our choice.

Referee 3's advocacy for linear dimensional reduction techniques overlooks the necessity and generality of non-linear approaches, as enabled by artificial deep neural network frameworks, demonstrated in the present manuscript. Nevertheless, our manuscript includes evidence demonstrating the optimality of our current reduced dimensions through varied dimensional analyses. Our extensive analysis, based on the VAMP-2 score, supports the sufficiency of the present dimensions compared to other linear reduction methods.

The referee’s views that developing Markov state models (MSM) of apo form of the alphasynulclein using multiple number of 1-4 microsecond long simulation length is misleading, suggests referee’s lack of knowledge on the fundamental purpose and motivation for the usage of MSM, which is, to derive long-time scale equilibrium properties from significantly short-length adaptively sampled trajectories. The referee has overlooked the extensive analysis that the authors had provided while demonstrating that the Markov state models developed from short length simulation trajectories of alpha-synclein can statistically replicate the properties derived from very long trajectories.

---

The following is the authors’ response to the original reviews.

The following extensive analyses were performed to address the reviewer comments:

(1) We have calculated the distribution of radius of gyration (Rg), end-to-end distance (Ree), solvent accessible surface area (SASA) of the apo and holo simulations and also their secondary structure composition.

(2) We have performed a similar analysis for the smaller 60 µs chunk from two parts of the 1.5 ms trajectory.

(3) The choice of starting structures have been thoroughly investigated in terms of Rg, RMSD, secondary structure and SASA.

(4) We have justified the use of VAE over tICA.

(5) We have verified the choice of hyperparameters that were used to build the MSM.

(6) We have retrained a VAE with four latent dimensions and used it to build MSM.

(7) As per recommendation of the referee 1, we have updated the title of the manuscript by introducing ‘expansion’ phrase.

The manuscript has been accordingly revised by updating it with additional analysis.

**Public Reviews:**

**Reviewer #1 (Public Review):**
Summary:This is a well-conducted study about the mechanism of binding of a small molecule (fasudil) to a disordered protein (alpha-synuclein). Since this type of interaction has puzzled researchers for the last two decades, the results presented are welcome as they offer relevant insight into the physical principles underlying this interaction.Strengths:The results show convincingly that the mechanism of entropic expansion can explain the previously reported binding of fasudil to alpha-synuclein. In this context, the analysis of the changes in the entropy of the protein and of water is highly relevant. The combination use of machine learning for dimensional reduction and of Markov State Models could become a general procedure for the analysis of other systems where a compound binds a disordered protein.Weaknesses:It would be important to underscore the computational nature of the results, since the experimental evidence that fasudil binds alpha-synuclein is not entirely clear, at least to my knowledge.

The experimental evidence of binding of fasudil to α-synuclein and potentially preventing its aggregation is reported in the paper “Fasudil attenuates aggregation of α-synuclein in models of Parkinson’s disease. Tatenhorst et al. Acta Neuropathologica Communications (2016) 4:39 DOI 10.1186/s40478-016-0310-y ”. In this work, solution state 15N-1H HSQC NMR experiments were performed of α-synuclein in increasing amounts of fasudil which led to large chemical shift perturbation of Y133 and Y136 residues. Additionally single and double mutant synT-Y133A and synT-Y136A (tyrosine is replaced with alanine), when treated with fasudil, had no significant effect as evident from immunochemistry, thereby indicating that α-synuclein aggregation can be inhibited by the interaction of C-terminal tyrosines with fasudil. These two analyses point to binding specific binding sites of fasudil to α-synuclein.

In our work, we have built a MSM using the latent dimension of a deep learning method called VAE, to address how fasudil interacts with α-synuclein. An analysis of the macrostates as obtained from MSM, gives insights into how fasudil interacts with α-synuclein, in terms of transition probabilities among the states, thereby predicting which states are most favorable for binding.

**Reviewer #2 (Public Review):**
The manuscript by Menon et al describes a set of simulations of alpha-Synuclein (aSYN) and analyses of these and previous simulations in the presence of a small molecule.While I agree with the authors that the questions addressed are interesting, I am not sure how much we learn from the present simulations and analyses. In parts, the manuscript reads more like an attempt to apply a whole range of tools rather than with a goal of answering any specific questions.

In this manuscript, we have employed a variational bayesian method, VAE, that uses variational inference to approximate the distribution of latent variable. Unlike conventional linear dimension reduction methods such as tICA (as provided in the SI), this method has been found to be better (higher VAMP2 score) in capturing slow modes and thereby facilitate the study of long-time dynamics. Markov State Model was built on this lower dimension space which indicated the presence of three and six states for the apo and holo simulations respectively. The exclusivity of the states was justified by determining the backbone contact map and further mapping these states using a denoising CNN-VAE. The increase in the number of states in the presence of the small molecule was justified by calculating the entropy of the macrostates. The entropic contribution from water remained similar across all states, while for the protein in the holo ensemble, entropy was significantly modulated (either increased or decreased) compared to the apo state. In contrast, the entropy of the apo states showed much less modulation. This proves that an increase in the number of states is primarily an entropic effect caused by the small molecule. Finally we have compared the mean first passage time (MFPT) of other states to the most populated state, which reveals a strong correlation between transition time and the system's entropy for both apo and holo ensemble. However, the transition times (to the most populated state) are much lower for the holo ensemble, thereby suggesting that fasudil may potentially trap the protein conformations in the intermediate states, thereby slowing down αS in exploring the large conformational space and eventually slow down aggregation.

There's a lot going on in this paper, and I am not sure it is useful for the authors, readers or me to spell out all of my comments in detail. But here are at least some points that I found confusing/etcMajor concernsp. 5 and elsewhere:I lack a serious discussion of convergence and the statistics of the differences between the two sets of simulations. On p. 5 it is described how the authors ran multiple simulations of the ligandfree system for a total of 62 µs; that is about 25 times less than for the ligand system. I acknowledge that running 1.5 ms is unfeasible, but at a bare minimum the authors should discuss and analyse the consequences for the relatively small amount of sampling. Here it is important to say that while 62 µs may sound like a lot it is probably not enough to sample the relevant properties of a 140-residue long disordered protein.

As to referee 2’s original comment on ‘a lot going on in the manuscript’, we believe that the complexity of the project demanded that this work needs to be dealt with an extensive analysis and objective machine learning approaches, instead of routine collective variable or traditional linear dimensional reduction techniques. This is what has been accomplished in this manuscript. For someone to get the gist of the work, the last paragraph of the introduction and first paragraph of conclusion provides a summary of the overall finding and investigation in the manuscript. First, a VAE-based machine learning approach demonstrates the modulation of free energy landscape of alpha-synuclein in presence of fasudil. Next, Markov State Model elucidates distinct binding competing states of alpha-synuclein in presence of the small-molecule drug. Then the MSMderived metastable states of alpha-synuclein monomer are structurally characterized in presence of fasudil. Next we mapped the macrostates in apo and bound-state ensembles using denoising convolutional variational autoencoder, to ensure that these are mutually distinct. Next we show that fasudil exhibits conformation-dependent interactions with individual metastable states. Finally the investigation quantatively brings out entropic signatures of small molecule binding.

We thank the reviewer for the question. For the apo simulations, we performed 1-4 μs long simulations with 23 different starting structures and the ensemble amounted to an ensemble of ~62 μs. In the Supplementary figures, we show analyses of how the starting structures used for apo simulations compare with the structure used to run the holo simulations as well as comparison of the apo and holo ensembles in terms of structures features as Rg, Ree, solvent accessible surface area (SASA) and secondary structure properties. This is updated in the manuscript on page 3,31- 33 and figures S1-S6, S25-S30.

Also, regarding the choice of starting structures, we chose multiple distinct conformations from a previous simulation of alpha synuclein monomer, reported in *Robustelli et. al, PNAS, 115 (21), E4758-E4766.* The Rg of the starting structures represent the entire distribution of Rg of the holo ensemble; from compact, intermediate to extended states. Importantly, the Rg distribution of the apo and holo ensembles are highly comparable and overlapping, indicating that the apo simulations, although of short timescale, have sampled the phase space locally around each starting conformation and thus covered the protein phase space as in the holo simulation. Similarly, other structural properties such as SASA, Ree and secondary structure are comparable for the two ensembles. These analyses show that the local sampling across a variety of starting conformations has ensured sufficient sampling of the IDP phase space. This is updated in the manuscript on page 33-34 and figure S1, S25-S30.

p. 7:The authors make it sound like a bad thing than some methods are deterministic. Why is that the case? What kind of uncertainty in the data do they mean? One can certainly have deterministic methods and still deal with uncertainty. Again, this seems like a somewhat ad hoc argument for the choice of the method used.

We appreciate the reviewer’s comment. In this work, we have used a single VAE model to map the simulation of αS in its apo state and in the presence of fasudil, into two dimensions. If we had used an autoencoder, which is a deterministic model, we would have to train two independent models; one for the apo-state and one for fasudil. It would then be questionable to compare the two dimensions obtained from two different autoencoders as the model parameters are not shared.

VAE gives us this flexibility by not mapping it to a single point, but to a distribution, thereby encouraging it to learn more generalizable representation. The uncertainty is not in the data; but mapping a conformation (of the fasudil simulation) to a distribution would provide a new point for a similar structure (from the apo simulation).

p. 8:The authors should make it clear (i) what the reconstruction loss and KL is calculated over and (ii) what the RMSD is calculated over.

(i) The reconstruction loss is calculated between the reconstructed and original pairwise distances, whereas the KL loss is calculated between the approximated posterior distribution and the prior distribution (for VAE it is a standard normal distribution)

(ii) The RMSE is the root mean square error between the original data and the reconstructed data.

(i) is updated on page 34 and (ii) is updated in the revised manuscript on page 8.

p. 9/figure 1:The authors select a beta value that may be the minimum, but then is just below a big jump in the cross-validation error. Why does the error jump so much and isn't it slightly dangerous to pick a value close to such a large jump.

In this work, RMSE has been chosen as a metric to select the best VAE model. To do so, the β parameter (weighting factor for the KL loss) was varied. The β value was chosen as this had the minimum value.

This is updated on page 8.

p. 10:Why was a 2-dimensional representation used in the VAE? What evidence do the authors have that the representation is meaningful? The authors state "The free energy landscape represents a large number of spatially close local minima representative of energetically competitive conformations inherent in αS" but they do not say what they mean by "spatially close". In the original space? If so, where is the evidence.

We thank the reviewer for the question. Even though an increase in the number of latent dimensions may make the model more accurate, this can also result in overfitting. The model can simply memorize the pattern in the data instead of generalizing them. A higher dimensional latent space is also more difficult to interpret; therefore, we chose two dimensions.

The reconstruction loss (which is the mean squared error between the input and the reconstructed data) is of the order of 10-4. Also, the MSM built on the latent space of VAE is able to identify states that are distinct for both apo and holo simulations, which ensures that the latent space representation is meaningful.

We have also trained a model with 4 neurons in the latent space and built an MSM. The implied timescales indicate the presence of six states which is consistent with the model with two latent dimensions.

This is updated in the manuscript on page 13 and figure S14-S15.

No, not spatially close in the original space, but in the reduced two dimensional latent space.

p. 10:It is not clear from the text whether the VAEs are the same for both aSYN and aSYN-Fasudil. I assume they are. Given that the Fasudil dataset is 25x larger, presumably the VAE is mostly driven by that system. Is the VAE an equally good representation of both systems?

Yes, the same model is used for both aSYN and aSYN-Fasudil ensemble.

The states obtained from the MSM of the aSyn ensemble are distinct when their Cα contact maps are analyzed. So we think it is a good representation for this system.

p. 10/11:Do the authors have any evidence that the latent space representation preserves relevant kinetic properties? This is a key point because the entire analysis is built on this. The choice of using z1 and z2 to build the MSM seems somewhat ad hoc. What does the auto-correlation functions of Z1 and Z2 look like? Are the related to dynamics of some key structural properties like Rg or transient helical structure.

Autocorrelation of z1 and z2 of the latent space of VAE and the radius of gyration for asyn-fasudil simulation.

**Author response image 5. sa3fig5:** 

We find that z1 of VAE has a much slower decay as compared to Rg. This indicates that it is much better in capturing long-time-scale dynamics as compared to Rg.

p. 11:What's the argument for not building an MSM with states shared for aSYN +- Fasudil?

We have built two different markov state models for two aSYN simulation in its apo state and in the presence of ligand. Mixing the two latent spaces to build one MSM would give incorrect transition timescales among the states as these are independent simulations.

p. 12:Fig. 3b/c show quite clearly that the implied timescales are not converged at the chosen lag time (incidentally, it would have been useful with showing the timescales in physical time). The CK test is stated to be validated with "reasonable accuracy", though it is unclear what that means.

We have mentioned the physical timescales in the main manuscript (Page no. 38), which is 36 and 32 ns for apo and holo simulations, respectively. We used “reasonable accuracy” in the context of the Chapman-Kolmogorov test. We note that for the ligand simulations, the estimated and predicted models are in excellent agreement as compared to some of the transitions in the apo state. This good agreement implies that the model has reached Markovianity and the timescales have converged.

The CK test is updated in the manuscript on page 12.

p. 12:In Fig. 3d, what are the authors bootstrapping over? What are the errors if the authors analyse sampling noise (e.g. bootstrap over simulation blocks)?

For bootstrapping, we randomly deleted a part of the simulation (simulation block) and rebuilt the MSM with this reduced dataset. We repeated this 10 times and reported the average value of the population and the transition timescales over the 10 iterations.

p. 13:I appreciate that the authors build an MSM using only a subset of the fasudil simulations. Here, it would be important that this analysis includes the entire workflow so that the VAE is also rebuilt from scratch. Is that the case?

The VAE model was trained over data points of the ligand simulation sampled at every 9 ns starting from time t=0, for the entire 1.5 ms. We did not train it for the subset of the fasudil simulation, but rather used the trained VAE model to get the latent space of the 60 μs of the fasudil simulation to build the MSM. Additionally, we have compared the distributions of Rg for this simulation block with the apo ensemble and found good agreement among them.

Rg distribution is updated in the manuscript on page 13 and see figure S10-S11.

p. 18:I don't understand the goal of building the CVAE and DCVAE. Am I correct that the authors are building a complex ML model using only 3/6 input images? What is the goal of this analysis. As it stands, it reads a bit like simply wanting to apply some ML method to the data. Incidentally, the table in Fig. 6C is somewhat intransparent.

We appreciate the reviewer’s valid question. The ensemble averaged contact map of the macrostates of aSyn in apo state and in the presence of ligand posed us a challenge in finding contacts that are exclusive to each state. Since VAEs are excellent in finding patterns, we employed a convolutional VAE (typically used for images). However, owing to the few number of contact maps, the model overfitted and to prevent this, we added noise to the data. A visual inspection of the ensemble averaged contact map, especially for IDPs is difficult and this lower dimensional space will give us a preliminary idea of how each macrostate is different from every other. The table in Fig. 6C provides scores for the denoised contact maps (SSIM and PSNR scores). An SSIM score above 0.9 and PSNR score between 20-48 indicates that the reconstruction of the contact map is of good quality.

p. 22:"Our results indicate that the interaction of fasudil with αS residues governs the structural features of the protein."What results indicate this?

By building a Markov State Model and comparing them across the apo and holo ensembles, we showed the interaction of fasudil with aSyn leads to the population of more states (than apo). In these states, we observe that fasudil interacts with aSyn in different regions as shown by the protein-ligand contact map as shown in figure 7. Also, the contact maps and the extent of secondary structure of the six states are distinct across the states. The location and extent of the helix and sheet-like character in the ensemble of the six macrostates as shown in figure S16-S17. Based on these observations, we state that the interaction of the small molecule favors the population of new aSyn states that are distinct in their structural features.

p. 23:The authors should add some (realistic) errors to the entropy values quoted. Fig. 8 have some error bars, though they seem unrealistically small. Also, is the water value quoted from the same force field and conditions as for the simulations?

The error values are the standard deviations that are provided by the PDB2ENTROPY package. Yes, the water value is from the same force field and conditions for the simulations are the same as reported in the section “Entropy of water”

p. 23:Has PDB2ENTROPY been validated for use with disordered proteins?

Yes, it has been used in the following paper studying liquid-liquid phase separation of an IDP.

This paper has also been cited in the manuscript (reference 66).

“Thermodynamic forces from protein and water govern condensate formation of an intrinsically disordered protein domain” by Saumyak Mukherjee & Lars V. Schäfer, *Nature Communications* volume 14, Article number: 5892 (2023) https://doi.org/10.1038/s41467-023-41586-y

p. 23/24:It would be useful to compare (i) the free energies of the states (from their populations), (ii) the entropies (as calculated) and (iii) the enthalpies (as calculated e.g. as the average force field energy). Do they match up?

Our analysis stems from previous studies where enthalpy driven drug design has not led to significant advances in drug design, particularly for IDPs. In the presence of the drug/ligand, the protein may be able to explore a larger conformational space and hence an increase in the number of states accessible by the protein, which we found by building Markov State Model using the latent space of VAE. The entropy of the protein is calculated based on the torsional degrees of freedom relative to the random distribution (the protein with the most random configuration).

p. 31:It is unclear which previous simulation the new aSYN simulations were launched from. What is the size of the box used?

The starting conformations for the new aSYN simulations were randomly chosen from a previously reported 73 μs simulation in Robustelli et. al. (PNAS, 115 (21), E4758-E4766).

Box size for the 23 simulation has been added to the supplemental information in Table S1.

**Reviewer #3 (Public Review):**
Summary:In this manuscript Menon, Adhikari, and Mondal analyze explicit solvent molecular dynamics (MD) computer simulations of the intrinsically disordered protein (IDP) alpha-synuclein in the presence and absence of a small molecule ligand, Fasudil, previously demonstrated to bind alpha-synuclein by NMR spectroscopy without inducing folding into more ordered structures. In order to provide insight into the binding mechanism of Fasudil the authors analyze an unbiased 1500us MD simulation of alpha-synuclein in the presence of Fasudil previously reported by Robustelli et.al. (Journal of the American Chemical Society, 144(6), pp.2501-2510). The authors compare this simulation to a very different set of apo simulations: 23 separate1-4us simulations of alphasynuclein seeded from different apo conformations taken from another previously reported by Robustelli et. al. (PNAS, 115 (21), E4758-E4766), for a total of ~62us.To analyze the conformational space of alpha-synuclein - the authors employ a variational autoencoder (VAE) to reduce the dimensionality of Ca-Ca pairwise distances to 2 dimensions, and use the latent space projection of the VAE to build Markov state Models. The authors utilize kmeans clustering to cluster the sampled states of alpha-synuclein in each condition into 180 microstates on the VAE latent space. They then coarse grain these 180 microstates into a 3macrostate model for apo alpha-synuclein and a 6-macrostate model for alpha-synuclein in the presence of fasudil using the PCCA+ course graining method. Few details are provided to explain the hyperparameters used for PCCA+ coarse graining and the rationale for selecting the final number of macrostates.The authors analyze the properties of each of the alpha-synuclein macrostates from their final MSMs - examining intramolecular contacts, secondary structure propensities, and in the case of alpha-synuclein:Fasudil holo simulations - the contact probabilities between Fasudil and alphasynuclein residues.The authors utilize an additional variational autoencoder (a denoising convolutional VAE) to compare denoised contact maps of each macrostate, and project onto an additional latent space. The authors conclude that their apo and holo simulations are sampling distinct regions of the conformational space of alpha-synuclein projected on the denoising convolutional VAE latent space.Finally, the authors calculate water entropy and protein conformational entropy for each microstate. To facilitate water entropy calculations - the author's take a single structure from each macrostate - and ran a 20ps simulation at a finer timestep (4 femtoseconds) using a previously published method (DoSPT), which computes thermodynamic properties of water from MD simulations using autocorrelation functions of water velocities. The authors report that water entropy calculated from these individual 20ps simulations is very similar.For each macrostate the authors compute protein conformational entropy using a previously published Maximum Information Spanning tree approach based on torsion angle distributions - and observe that the estimated protein conformational entropy is substantially more negative for the macrostates of the holo ensemble.The authors calculate mean first passage times from their Markov state models and report a strong correlation between the protein conformational entropy of each state and the mean first passage time from each state to the highest populated state.As the authors observe the conformational entropy estimated from macrostates of the holo alphasynuclein:Fasudil is greater than those estimated from macrostates of the apo holo alphasynuclein macrostates - they suggest that the driving force of Fasudil binding is an increase in the conformational entropy of alpha-synuclein. No consideration/quantification of the enthalpy of alpha-synuclein Fasudil binding is presented.Strengths:The author's utilize MD simulations run with an appropriate force field for IDPs a99SB-disp and a99SB-disp water (Robustelli et. al, PNAS, 115 (21), E4758-E4766) - which has previously been used to perform MD simulations of alpha-synuclein that have been validated with extensive NMR data.The contact probability between Fasudil and each alpha-synuclein residue observed in the previously performed 1500us MD simulation of alpha-synuclein in the presence of Fasudil (Robustelli et. al., Journal of the American Chemical Society, 144(6), pp.2501-2510) was previously found to be in good agreement with experimental NMR chemical shift perturbations upon Fasudil binding - suggesting that this simulation is a reasonable choice for understanding IDP:small molecule interactions.Weaknesses:Major Weakness 1: Simulations of apo alpha-synuclein and holo simulations of alpha-synuclein and fasudil are not comparable.The most robust way to determine how presence of Fasudil affects the conformational ensemble of alpha-synuclein conclusions is to run apo and holo simulations of the same length from the same starting structures using the same simulation parameters.The 23 1-4 us independent simulations of apo alpha-synuclein and the long unbiased 1500us alpha-synuclein in the presence of fasudil are not directly comparable. The starting structures of simulations used to build a Markov state model to describe apo alpha-synuclein were taken from a previously reported 73us MD simulation of alpha-synuclein run with the a99SB-disp force field and water model with 100mM NaCl, (Robustelli et. al, PNAS, 115 (21), E4758-E4766). As the holo simulation of alpha-synuclein and Fasudil was run in 50mM NaCl, snapshots from the original apo alpha-synuclein simulation were resolvated with 50mM NaCl - and new simulations were run.No justification is offered for how starting structures were selected. We have no sense of the conformational variability of the starting structures selected and no sense of how these conformations compare to the alpha-synuclein conformations sampled in the holo simulation in terms of standard structural descriptors such as tertiary contacts, secondary structure, radius of gyration (Rg), solvent exposed surface area etc. (we only see a comparison of projections on an uninterpretable non-linear latent-space and average contact maps). Additionally, 1-4 us is a relatively short timescale for a simulation of a 140 residue IDP- and one is unlikely to see substantial evolution for many structural properties of interest (ie. secondary structure, radius of gyration, tertiary contacts) in simulations this short. Without any information about the conformational space sample in the 23 apo simulations (aside from a projection on an uninterpretable latent space)- we have no way to determine if we observe transitions between distinct states in these short simulations, and therefore if it is possible the construct a meaningful MSM from these simulations.If the structures used for apo simulations are on average more compact or contain more tertiary contacts - then it is unsurprising that in short independent simulations they sample a smaller region of conformational space. Similarly, if the starting structures have similar dimensions - but we only observe extremely local sampling around starting structures in apo simulations in the short simulation times - it would also not be surprising that we sample a smaller amount of conformational space. By only presenting comparisons of conformational states on an uninformative VAE latent space - it is not possible for a reader to ask simple questions about how the conformational ensembles compare.It is noted that the authors attempt to address questions about sampling by building an MSM of single contiguous 60us portion of the holo simulation of alpha-synuclein and Fasudil - noting that:"the MSM built using lesser data (and same amount of data as in water) also indicated the presence of six states of alphaS in presence of fasudil, as was observed in the MSM of the full trajectory. Together, this exercise invalidates the sampling argument and suggests that the increase in the number of metastable macrostates of alphaS in fasudil solution relative to that in water is a direct outcome of the interaction of alphaS with the small molecule."However, the authors present no data to support this assertion - and readers have no sense of how the conformational space sampled in this portion of the trajectory compares to the conformational space sampled in the independent apo simulations or the full holo simulation. As the analyzed 60us portion of the holo trajectory may have no overlap with conformational space sampled in the independent apo simulations - it is unclear if this control provides any information. There is no quantification of the conformational entropy of the 6 states obtained from this portion of the holo trajectory or the full conformational space sampled. No information is presented to determine if we observe similar states in the shorter portion of the holo trajectory. Furthermore - as the authors provide almost no justification for the criteria used to select of the final number of macrostates for any of the MSMs reported in this work- and the number of macrostates is effectively a free parameter in the PCCA+ method, arriving at an MSM with 6 macrostates does not convey any information about the conformational entropy of alpha-synuclein in the presence or absence of ligands. Indeed - the implied timescale plot for 60us holo MSM (Figure S2) - shows that at least 10 processes are resolved in the 120 microstate model - and there is no information to provided explaining/justifying how a final 6-macrostate model was determined. The authors also do not project the conformations sampled in this sub- trajectory onto the latent space of the final VAE.One certainly expects that an MSM built with 1/20th of the simulation data should have substantial differences from an MSM built from the full trajectory - so failing additional information and hyperparameter justification - one wonders if the emergence of a 6-state model could be the direct result of hardcoded VAE and MSM construction hyperparameter choices.Required Controls For Supporting the Conclusions of the Study: The authors should initiate apo and holo simulations from the same starting structures - using the same simulation software and parameters. This could be done by adding a Fasudil ligand to the apo structures - or by removing the Fasudil ligand from a subset of holo structures. This would enable them to make apples-toapples comparisons about the effect of Fasudil on alpha-synuclein conformational space.Failing to add direct apples-to-apples comparisons, which would be required to truly support the studies conclusions, the authors should at least compare the conformational space sampled in the independent apo simulations and holo simulations using standard interpretable IDP order parameters (ie. Rg, end-to-end distance, secondary structure order parameters) and/or principal components from PCA or tICA obtained from the holo simulation. The authors should quantify the number of transitions observed between conformational states in their apo simulations. The authors could also perform more appropriate holo controls, without additional calculations, by taking batches of a similar number of short 1-4us segments of simulations used to compute the apo MSMs and examining how the parameters/macrostates of the holo MSMs vary with the input with random selections.

In case of IDPs, one should not bias the simulation by starting from identical structures, as IDP does not have a defined structure and the starting configuration has little significance. It is the microenvironment that matters most. As for the choice of simulation software and parameters, we have used the same force field that was used in the holo simulation at the same temperature and same salt concentration. We have performed multiple independent simulations that have varying structural signatures such as Rg, SASA and secondary structure content. In fact, the starting structure for apo simulations covered the entire span of the Rg distribution of holo simulation, including the starting structure of the holo simulation. The simulations are unbiased w.r.t the starting structure. Although the fasudil simulation was run for 1.5 ms, we should also understand that it is difficult to run a millisecond range of simulation in reasonable time from a single starting structure. It is exactly for this reason that we start with different structures so that we do not bias ourselves and sample every possible conformation.

We have updated the manuscript on page 33-34 and figure S1, S25-S30.

Considering the computational expense for simulating 1.5 ms timescale of a 140-residue IDP, we generated an ensemble from multiple short runs amounting to ~60 µs. The premise of this investigation is a widely popular method, Markov State Models (MSMs) that can be used to estimate long timescale kinetics and stationary populations of metastable states built from ensembles of short simulations. We have also demonstrated that comparable to the apo data, when we build an MSM for asyn-fasudil (holo) using 60 µs simulation block, the implied timescales (ITS) plot shows identical number of metastable states as for the 1.5 ms data.

An intrinsically disordered protein (IDP) is not represented by a fixed structure. Therefore, it would be most appropriate to run multiple simulations starting from different initial structures and simulate the local environment around those structures; thus generating an ensemble effectively sampling the phase space. Accordingly, for initiating the apo simulations, instead of biasing the initial structure (using the starting structure used for simulations with fasudil), we chose randomly 23 different conformations from the 73 µs long simulation of 𝛼-synuclein monomer reported in *Robustelli et. al, PNAS, 115 (21), E4758-E4766*. Based on the reviewer’s comment on providing a justification for choice of the starting structures for apo simulations, we provide a compilation of figures below showing comparison of standard conformational properties of the chosen initial structures for apo simulations with the starting structure of the long holo simulation; we have also provided comparative analyses of the apo (~60 µs) and holo ensemble (1.5 ms) properties.

Figure S1 compares the Rg of the apo and holo ensembles of ~60 μs and 1.5 ms, respectively. The distributions are majorly overlapping, indicating that the apo ensemble is comparable to the holo ensemble, in terms of the extent of compaction of the conformations. In Figure 1, we have also marked the Rg values corresponding to the starting structures used to seed the apo simulations. It is evident that the 23 starting conformations chosen represent the whole range of the Rg space that is sampled in the holo ensemble. Therefore, while the apo simulations are relatively short (1-4 μs), the local sampling of these multiple starting conformations of variable compaction (Rg) ensures that the phase space is efficiently sampled and the resulting ensemble is comparable to the holo ensemble. Furthermore, the implementation of MSM on such an ensemble can be efficiently used to identify metastable states and the long timescale transitions happening between them

Another property that is proportional to Rg is the end-to-end distance of the protein conformations. Figure S2 shows that the distribution of this property in the apo and holo ensembles are highly similar.

Figure S3 depicts another fundamental structural descriptor i.e. solvent accessible surface area (SASA) that indicates the extent of folding and the exposure of the residues. The apo ensemble only shows a minimal shift in the distribution towards higher SASA values. The distributions of the two ensembles largely overlap.

In Figure S25, we have provided the root mean square deviation (RMSD) of the starting structures used in the apo simulations with the structure used to start the long simulation with fasudil. The RMSD values range from 1.6 to 3 nm, indicating that the starting structures used are highly variable. This is justifiable for IDPs since they are not identified by a single, fixed structure, but rather by an array of different conformations.

Figures S26-S28 show the fraction of the secondary structure elements i.e. helix, beta and coil in the starting structures of apo and holo simulations. All the conformations are mostly disordered in nature with the greatest extent of coil content. The helix content ranges from 3-10 % while sheet content varies from 3-15 % in the initial simulation structures.

Figures S4-s6 represent the residue-wise percentage of secondary structure elements (helix, beta and coil) in the apo and holo ensembles. It is evident that the extent of secondary structure is comparable in the two ensembles.

The above analyses comparing distributions of several structural features clearly indicate that the apo simulations we performed from different starting structures have effectively sampled the phase space as the single long simulation of the holo system.

We have discussed the above in the manuscript: Computational Methods section, Page 33-34.

The above VAMP score analyses Figures S7 and S8has been now presented in the manuscript: Results and Discussion (Page 8)

Building the MSM

While building the MSM, we iteratively varied the hyperparameters to build a reasonable model. In this process, we explored different values of the number of clusters, maximum number of iterations, tolerance, stride, metric, seed, chunk size and initialization methods. There is no possible way to perform an optimization on the choice of the above hyperparameters using gradient descent methods, as no convergence would be guaranteed. The parameters were tuned carefully so that we get the best possible implied timescales of the system. The quality of the MSM was further validated using the Chapman-Kolmogorov (CK) test on a state-by-state basis i.e by considering the transitions between each pair of the metastable states. In addition, we have built the contact maps to show that the states are mutually exclusive. This is also justified by the latent space of denoising convolutional variational autoencoders.

We have compared the conformational space in the independent apo and holo simulations for Rg, Ree, SASA and secondary structure. As for PCA/TICA, we have computed the VAMP-2 score for TICA and found out to be low as compared to VAE. In fact, neural networks have been shown previously as a better dimension reduction technique due to its non-linearity over linear methods such as PCA or TICA.

**Author response image 6. sa3fig6:** Distribution of (a)Rg, (b) Ree, (c) SASA and of the apo ensemble and a 60 μs slice of the holo simulation trajectory. (d) ITS plot of the 60 μs chunk.

First, someone familiar with MSM should understand that the basic philosophy of MSM is not the requirement of long simulation trajectories, which would defeat the purpose of its usage. Rather as motivated by Noe and coworkers in seminal PNAS (vol. 106, page 9011, year 2009) paper, MSM plays an important role in inferring long-time scale equilibrium properties by using significantly short-length scale non-equilibrium trajectories.

Considering the difference in the size of the ensembles in the apo and holo simulations, we verified how different is the MSM built using 60 μs slice of the data from the 1.5 ms holo simulation in terms of the number of metastable states identified by the model. For this, we considered 60 μs data beginning from 966 μs - 1026 μs. First, we compared the gross structural properties of these datasets. Author response image 6a-c compares the distributions of Rg, Ree and SASA. The distributions show that the apo and holo simulations are very similar with respect to these standard properties of protein conformations.

We built the MSM for this 60 μs data of the holo ensemble from the reduced data obtained from the same VAE model. We would like to clarify that the hyperparameters of the model are not hardcoded but rather carefully fine-tuned to obtain a good model that performs good kinetic discretization of the underlying macrostates. The implied timescale plot of this new MSM shows distinct timescales corresponding to six macrostates. This led us to conclude that the six-state model is robust despite the differences in the ensemble size. The implied timescale is shown in Author response image 6d.

The above analyses in Author response image 6 are presented in Results and Discussion, Page 13.

Major Weakness 2: There is little justification of how the hyperparameters MSMs were selected. It is unclear if the results of the study depend on arbitrary hyperparameter selections such as the final number of macrostates in each model.It is unclear what criteria were used to determine the appropriate number of microstates and macrostates for each MSM. Most importantly - as all analyses of water entropy and conformational entropy are restricted to the final macrostates - the criteria used to select the final number of macrostates with the PCCA+ are extremely important to the results of the conclusions of the study. From examining the ITS plots in Figure 3 - it seems both MSMs show the same number of resolved processes (at least 11) - suggesting that a 10-state model could be apropraite for both systems. If one were to simply select a large number of macrostates for the 20x longer holo simulation - do these states converge to the same conformational entropy as the states seen in the short apo simulations? Is there some MSM quality metric used to determine what number of macrostates is more appropriate?Required Controls For Supporting the Conclusions of the Study: The authors should specify the criteria used to determine the appropriate number of microstates and macrostates for their MSMs and present controls that demonstrate that the conformational entropies calculated for their final states are not simply a function of the ratio of the number macrostates chosen to represent very disparate amounts of conformational sampling.

VAMP-2 score was used to determine the number of microstates. We have calculated the VAMP2 score by varying the number of microstates, ranging from 10 to 220. We find that the VAMP-2 score has saturated at a higher number of microstates for both apo and holo simulations.

The number of macrostates were determined by the gap between the lines of the Implied timescales plot followed by a CK test (shown in figure S1). Since we plotted the first 10 slowest timescales, the implied timescales show 10 timescales and this is not an indicator of the number of macrostates. The macrostates are separated by distinct gaps in the timescales and do not merge as seen beyond 5 timescales in the plot. The timescales, when leveled off and distinct, indicate that the system has well defined metastable states and the MSM is accurate in identifying the macrostates. We find this to be three and six for the apo and holo simulations from the corresponding implied timescales.

The above is discussed in Computational Methods, Page 37-38.

Major Weakness 3: The use of variational autoencoders (VAEs) obscures insights into the underlying conformational ensembles of apo and holo alpha-synuclein rather than providing new onesNo rationale is offered for the selection of the VAE architecture or hyperparameters used to reduce the dimensionality of alpha-synuclein conformational space.It is not clear the VAEs employed in this study are providing any new insight into the conformational ensembles and binding mechanisms of Fasudil to alpha-synuclein, or if the underlying latent space of the VAEs are more informative or kinetically meaningful than standard linear dimensionality reduction techniques like PCA and tICA. The initial VAE is used to reduce the dimensionality of alpha-synuclein conformational ensembles to 2 degrees of freedom - but it is unclear if this projection is structurally or kinetically meaningful. It is not clear why the authors choice to use a 2-dimeinsional projection instead of a higher number of dimensions to build their MSMs. Can they produce a more kinetically and structurally meaningful model using a higher dimensional VAE latent space?Additionally - it is not clear what insights are provided by the Denoising Convolutional Variational Autoencoder. The authors appear to be noising-and-denoising the contact maps of each macrostate, and then projecting the denoised values onto a new latent space - and commenting that they are different. Does this provide additional insight that looking at the contact maps in Figures 4&5 does not? Is this more informative than examining the distribution of the Radii of gyration or the secondary structure propensities of each ensemble? It is not clear what insight this analysis adds to the manuscript.Suggested controls to improve the study: The authors should project interpretable IDP structural descriptors (ie. secondary structure, radius of gyration, secondary structure content, # of intramolecular contacts, # of intermolecular contacts between alpha-synuclein and Fasudil) onto this latent space to illustrate if any of these properties are meaningful separated by the VAE projection. The authors should compare these projections, and MSMs built from these projections, to projections and MSMs built from projections using standard linear dimensionality projection techniques like PCA and tICA.

We have already pointed out the IDP structural parameters for the first question.

In case of VAE, the latent space captures the underlying pattern of the higher dimensional data. A non-linear projection using VAE has shown to have a higher VAMP-2 score over linear dimension reduction methods such as tICA. The latent space of VAE was then used to build the MSM, in order to get the macrostates and also the transition timescales among them. We can project the data onto a higher dimension, but the goal is to reduce it to lower dimensions where it will be easier to interpret. Higher number dimensions would also risk overfitting; and the model, instead of learning the pattern, it may simply memorize the data. The training and validation loss curve from VAE has reached the order of 10^-4 thereby indicating good reconstruction of the original data.

As for dimension reduction using tICA, the VAMP-2 score confirms that our VAE model performs better than tICA. This manuscript uses deep neural networks to understand the structural and kinetic process of IDP and small molecule interaction. Dimension reduction using tICA would give different reaction coordinates and MSM built using the projected data of tICA will not be one-to one comparable with that obtained from VAE.

We had to perform noising, as we had only 9 contact maps. This led to overfitting of the CVAE model. To overcome this problem, we have introduced white noise to our data, so as to prevent the model from overfitting. The objective of the DCVAE model was to see how distinct these contact maps are based on their locations on a lower dimensional space. A visual inspection of the ensemble averaged contact map, especially for IDPs is much more difficult as compared to folded proteins. So, even before computing the Rg, Ree, SASA or secondary structure, this lower dimensional space will give us a preliminary idea of how each macrostate is different from every other.

As for the distribution of Rg, we have plotted it in Author response image 7. The residue-wise percentage secondary structure is plotted in figure S4-S6 for the holo and apo simulation respectively.

**Author response image 7. sa3fig7:** Distribution of radius of gyration for the three and six macrostates in the apo and holo simulation respectively.

As for training a model with a higher number of latent dimensions, we have retrained a VAE model with four dimensions in the latent space. The loss was of the order of 10-4. We built a MSM with the appropriate number of microstates and found the presence of six macrostates as evident from the ITS plot as shown in Figure S14 and S15.

This data is presented in Results and Discussion, Page 13

Major Weakness 4: The MSMs produced in this study have large discrepancies with MSMs previously produced on the same dataset by the same authors that are not discussed.Previously - two of the authors of this manuscript (Menon and Mondal) authored a preprint titled "Small molecule modulates α-synuclein conformation and its oligomerization via Entropy Expansion" (https://www.biorxiv.org/content/10.1101/2022.10.20.513005v1.full) that analyzed the same 1500us holo simulation of alpha-synuclein binding Fasudil. In this study - they utilized the variational approach to Markov processes (VAMP) to build an MSM using a 1D order parameter as input (the radius of gyration), first discretizing the conformational space into 300 microstates before similarly building a 6 macrostate model. From examining the contact maps and secondary structure propensities of the holo MSMs from the current study and the previous study- some of the macrostates appear similar, however there appear to be orders of magnitude differences in the timescales of conformational transitions between the two models. The timescales of conformational transitions in the previous MSM are on the order of 10s of microseconds, while the timescales of transitions in this manuscript are 100s-1000s microseconds. In the previous manuscript, a 3 state MSM is built from an apo α-synuclein obtained from a continuous 73ms unbiased MD simulation of alpha-synuclein run at a different salt concentration (100mM) and an additional 33 ms of shorter simulations. The apo MSM from the previous study similarly reports very fast timescales of transitions between apo states (on the order ~1ms) - while the MSM reported in the current study (Figure 9) are on the order of 10s-100s of microseconds.These discrepancies raise further concerns that the properties of the MSMs built on these systems are extremely sensitive to the chosen projection methods and MSM modeling choices and hyperparameters, and that neither model may be an accurate description of the true underlying dynamicsSuggestions to improve the study: The authors should discuss the discrepancies with the MSMs reported in their previous studies.

In the previous preprint, the radius of gyration was used as the collective variable to build the MSM. In this manuscript, we have used a much more general collective variable, reduced pairwise distance using VAE. Firstly, the collective variables used to build the model in the two works are different. Secondly, for the 73 μs apo simulation in the previous manuscript, the salt concentration used was 100 mM, but in this work, we have used a salt concentration of 50 mM, same as the salt concentration used in the holo simulations. Since the two simulation conditions are different with respect to salt concentration, the conformational space sampled in these conditions will be different and this will be reflected in the nature/features of the metastable states and the associated transition kinetics. Thirdly, the lag time at which the MSM was built was 3.6 ns in the previous manuscript, whereas, in this work we have used 32 ns. This is already off by a factor of 10. So the order of timescales have also changed. Thus, changes in the collective variable and change in the lag time at which the system reaches Markovianity is different. Hence, the timescales of transition among the macrostates are also different. Because of these differences, it would not be correct to compare the results that we would get from the two investigations.

**Recommendations for the authors:**

**Reviewer #1 (Recommendations For The Authors):**
To highlight the role of the entropic expansion mechanism, I would suggest modifying the title to capture this result, for example: "An Integrated Machine Learning Approach Delineates an Entropic Expansion Mechanism for the Binding of a Small Molecule to α-Synuclein".

We have changed the title as suggested by the reviewer.

To my knowledge the binding of fasudil to alpha-synuclein has been shown in the simulations by Robustelli et al (JACS 2022), but the experimental evidence is less clear cut. If an experimental binding affinity and the effect on alpha-synuclein aggregation have been measured, they should be reported.
**Reviewer #2 (Recommendations For The Authors):**

We thank the reviewer for the careful evaluation of our manuscript and providing comments and questions that we have attempted to address and incorporate.

MinorAbstract:In "which is able to statistically distinguish fuzzy ensemble", what does the word "statistically" mean in this context? Do the authors present evidence that the two ensembles are statistically different, and if so in what ways?

We have analyzed the apo and holo ensembles of aSyn using the framework of Markov State Models, which provides the stationary populations of the states that the model identifies. For this reason, we have used ‘which is able to statistically distinguish fuzzy ensemble’ as we compare and contrast the metastable states that we resolve using MSM. The MSM provides metastable states which are identified through statistical analysis of the transitions between states (transition probability matrix). We characterize their structural features to distinguish them which gives a meaningful interpretation of the fuzzy ensemble.

Abstract:What does "entropic ordering" mean?

We thank the reviewer for pointing this out. Here, we mean that the presence of the small molecule only affects the protein backbone entropy while the entropy of water is not affected in the simulations with fasudil. We will rewrite this more clearly in the abstract.

The changed sentence is as follows:

“A thermodynamic analysis indicates that small-molecule modulates the structural repertoire of αS by tuning protein backbone entropy, however the entropy of the water remains unperturbed.”

Abstract:What does "offering insights into entropic modulation" mean?

In this investigation, we first discretized the ensemble of a small-molecule binding/interacting with a disordered aSyn into the underlying metastable states, followed by characterisation of these identified states. As small molecule interactions can affect the overall entropy of the IDP, we estimated the said effect of fasudil binding on aSyn. We find that small molecule binding effect is manifested in the protein backbone entropy and the solvent entropy is not affected. Through this work, we highlight these insights into the modulatory effect that fasudil brings about in the entropy of the system (*entropic modulation*).

p. 3/4:When the authors write "However, a routine comparison of monomeric αS ensemble... ensemble" it is unclear whether they are referring to previous work they only cite a paper with simulations of "apo" aSYN, and if so which. Do they mean Ref 32? Also, the word "routine" sounds odd in this context.

We thank the author for pointing this out. We compared the ensemble properties (such as the distributions of the radius of gyration, end-to-end distance, solvent accessible surface area, secondary structure properties) of ɑ-synuclein monomer that we generated in neat water and the ensemble of ɑ-synuclein in the presence of the small molecule fasudil that is reported in Robustelli et.al. (Journal of the American Chemical Society, 144(6), pp.2501-2510). We have now modified this sentence in the main manuscript as follows: (Page no 3)

“However, comparison of the global and local structural features of the αS ensemble in neat water and that in the presence of fasudil [32] (see Figure S1-S6) did not indicate a significant difference that is a customary signature of the dynamic IDP ensemble.”

p. 4:Regarding "Integrative approaches are therefore gaining importance in IDP studies", these kinds of integrative approaches have been used for 20 years for studies of IDPs (with increasing sophistication and success), so I think "gaining" is somewhat of a stretch.

We thank the reviewer for this comment. We agree with the reviewer and have now changed this sentence as follows:

“Integrative approaches have been exploited in studying IDPs as well as small-molecule binding to IDPs.”

p. 5:What does "large scale" mean in "This study showed no large-scale differences between the bound and unbound states of αS"? Do the authors mean substantially/significantly different, or differences on a large (length) scale?

Here, we refer to the study of small molecule (fasudil) binding study to α-synclein reported in Robustelli et.al. (Journal of the American Chemical Society, 144(6), pp.2501-2510). In this study, the authors report no substantial (“large scale”) differences in the conformational ensembles of αsynuclein in the bound and unbound states of fasudil such as the backbone conformation distributions.

p. 6:The authors write "In a clear departure from the classical view of ligand binding to a folded globular protein, the visual change in αS ensemble due to the presence of small molecule is not so strikingly apparent." I don't understand this. Normally, there is very little difference between apo and holo protein structures for folded proteins, so I don't understand the "in a clear departure" part. This seems like a strawman. Of course, for folded proteins one can generally see the ligand bound, but here the authors are talking about the protein.

In case of folded proteins, the overall tertiary structure of the protein remains mostly the same upon binding of the ligand. Structural changes are localized in nature and primarily around the binding site. However, in case of ⍺Syn, binding of fasudil is transient and not as strong as seen for folded proteins. “Clear departure” refers to the fact that for ⍺Syn, binding of fasudil is more subtle and dispersed across the ensemble of conformations rather than localized changes as in case of folded proteins.

p. 6:I don't think the term "data-agnostic" makes sense since these methods are based on data and also make some assumptions about how the data can/should be used.

We have replaced this term with “model-agnostic”.

p. 16:How are contacts defined; please add to caption.

A contact is considered if the Cα atoms of two residues are within a distance of 8 Å of each other. We have updated the caption with this information in Figures 4 and 5.

p. 20:What do the authors mean by "non-specific interactions" in this context?

The interactions of fasudil are predominantly with the negatively charged residues in the C-terminal region of ⍺Syn via charge-charge and π-stacking interactions (Robustelli et.al. (Journal of the American Chemical Society, 144(6), pp.2501-2510)).

In addition, in some metastable states that we identify, we also observe transient interactions with residues in the hydrophobic NAC region and N-terminal region. We refer to these transient interactions as “non-specific” interactions.

p. 27:Are the axes of Fig. 9c/d z1 and z2?

Yes. The axes are z1 and z2

Smaller than minorAbstract:Rephrase "In particular, the presence of fasudil in milieu"

We have rephrased the sentence as follows:

“In particular, the presence of fasudil in the solvent…”

p. 4:What does the word "potentially" do in "ensemble of conformations potentially sampled"?

Here, by potentially, we mean the various conformations that the protein can adopt, subject to the environmental conditions.

p. 10:"we trained a large array of inter-residue pairwise distances"The distances were not trained; please reformulate

We have corrected this sentence as follows:

“We trained a VAE model using a large array of inter-residue pairwise distances.”

p. 13:N/C-terminal -> terminus (or in the C-terminal region)

We have made the changes in the manuscript at the required places.

p. 20:Precedent -> previous (?)

We have made the change in the manuscript.

p. 30:As far as I understand, Anton does not use GPUs and does not run Desmond.

We thank the reviewer for providing this information. We referred to the original paper of the ⍺syn-fasudil simulations (Robustelli et.al. (Journal of the American Chemical Society, 144(6), pp.2501-2510)). The authors have performed equilibration with GPU/Desmond and used Anton for production runs. We have modified this sentence as:

We have modified this sentence as:

“A 1500 μs long all-atom MD simulation trajectory of αS monomer in aqueous fasudil solution was simulated by D. E. Shaw Research with the Anton supercomputer that is specially purposed for running long-time-scale simulations.” on page 31

References :

(1) Schütte C, Fischer A, Huisinga W, Deuflhard P (1999) A direct approach to conformational dynamics based on hybrid monte carlo. J Comput Phys 151:146–168

(2) Chodera JD, Swope WC, Pitera JW, Dill KA (2006) Long-time protein folding dynamics from short-time molecular dynamics simulations.Multiscale Model Simul5:1214–1226.